# Heritable polygenic editing: the next frontier in genomic medicine?

Peter M. Visscher[1,2 ✉], Christopher Gyngell[3,4], Loic Yengo[1] & Julian Savulescu[3,5,6 ✉]

Polygenic genome editing in human embryos and germ cells is predicted to become feasible in the next three decades. Several recent books and academic papers have outlined the ethical concerns raised by germline genome editing and the opportunities that it may present[1–3]. To date, no attempts have been made to predict the consequences of altering specific variants associated with polygenic diseases. In this Analysis, we show that polygenic genome editing could theoretically yield extreme reductions in disease susceptibility. For example, editing a relatively small number of genomic variants could make a substantial difference to an individual's risk of developing coronary artery disease, Alzheimer's disease, major depressive disorder, diabetes and schizophrenia. Similarly, large changes in risk factors, such as low-density lipoprotein cholesterol and blood pressure, could, in theory, be achieved by polygenic editing. Although heritable polygenic editing (HPE) is still speculative, we completed calculations to discuss the underlying ethical issues. Our modelling demonstrates how the putatively positive consequences of gene editing at an individual level may deepen health inequalities. Further, as single or multiple gene variants can increase the risk of some diseases while decreasing that of others, HPE raises ethical challenges related to pleiotropy and genetic diversity. We conclude by arguing for a collectivist perspective on the ethical issues raised by HPE, which accounts for its effects on individuals, their families, communities and society[4].

In 2018, He Jiankui announced the birth of two babies, Lulu and Nana, whose genomes were edited in an attempt to make them immune to human immunodeficiency virus[1]. This has led to international outrage, the imprisonment of He Jiankui and numerous calls for a moratorium on reproductive gene editing[5]. Notably, He Jiankui was working outside of national regulations and international consensus on gene editing and other embryonic research and was breaching more general principles of research ethics, such as the requirement for informed consent[3]. Nevertheless, this scandal highlights the relatively advanced (although still error-prone) status of gene editing technologies and the need for a translational pathway if this technology is to be used in humans[6–8]. The birth of Lulu and Nana was followed by the birth of Aurea—the first child born via embryo screening using polygenic scores (ESPS[9]).

In recent decades, genetic studies in human populations have led to the discovery of tens of thousands of DNA variants associated with one or more complex traits, including common diseases such as autoimmune diseases, diabetes, heart disease, cancer and psychiatric disorders, as well as quantitative traits such as blood pressure, body mass index and height[10]. In isolation, trait-associated variants tend to have very small effects (less than around 1% of the trait standard deviation). However, the cumulative effect size across loci can be substantial. For complex traits, the effect of a polygenic score (the sum of risk variants across multiple loci weighted by the estimated effect size on risk) is comparable to that of known Mendelian mutations[11]. For example, for human height, the effect size of common alleles at height-associated loci is approximately 1 mm (around 1.5% of the phenotypic standard deviation) or less, but the standard deviation of a polygenic predictor based on approximately 12,000 genome-wide significant (GWS) loci is more than 40 times larger at around 4 cm (ref. 12).

Sample sizes from genome-wide association studies (GWAS) are increasing and will lead to larger effect sizes of polygenic scores. This is expected to underpin the greater efficacy of ESPS in the future. Nevertheless, it is currently not possible to use embryo selection on polygenic score to achieve large-scale changes in polygenic conditions[9,13]. Theoretical calculations imply that tens of thousands of embryos would be needed per couple to achieve a one standard deviation change in phenotype[13], which is infeasible and unlikely to gain social acceptance or ethical approval[14].

Gene editing technologies potentially allow germline editing of multiple targeted loci. In principle, these loci could be those identified from genetic association studies. Currently, very few causal variants for common disease are known with certainty. This is likely to change within a generation because of larger sample sizes, increased genome coverage and improved functional annotation. GWAS conducted in increasingly larger and genetically diverse samples and with increased genome coverage have a better chance of identifying variants that are causal[15], and functional annotation aids fine-mapping[16]. Furthermore, it is possible to test the functional effects of variants on protein

[1]Institute for Molecular Bioscience, University of Queensland, Brisbane, Australia. [2]Big Data Institute, Li Ka Shing Centre for Health Information and Discovery, Nuffield Department of Population Health, University of Oxford, Oxford, UK. [3]Murdoch Children's Research Institute, Melbourne, Victoria, Australia. [4]Department of Paediatrics, University of Melbourne, Melbourne, Australia. [5]Uehiro Oxford Institute, University of Oxford, Oxford, UK. [6]Centre for Biomedical Ethics, Yong Loo Lin School of Medicine, National University of Singapore, Singapore, Singapore. ✉e-mail: peter.visscher@uq.edu.au; julian.savulescu@philosophy.ox.ac.uk

expression in vitro using tools, such as experimental genome editing. Although it is not currently possible to target hundreds or thousands of polymorphisms simultaneously using gene editing, the rapid development of gene editing technology (for example, CRISPR–Cas9 gene editing was first reported in 2012), including advances in multiplex gene editing[17–19], leads us to believe that we might be one human generation (about 30 years) away from it becoming technically possible to perform gene editing on tens or hundreds of variants that are causal for common diseases. Whether multiplex editing of polygenic traits becomes practical or desirable, given the balance of risks and benefits, is highly uncertain and will depend, in part, on unknown safety and efficiency considerations. It is a prospect that is worth taking seriously, given the potentially disruptive and transformative consequences. The social and ethical structures that will need to be established, should heritable polygenic editing (HPE) becomes available, will have a profound impact on future generations. Although genetic engineering in humans has been discussed for decades, it has predominantly been discussed in the abstract.

We are now poised to frame an ethical discussion on the possible consequences of changing specific variants linked to complex diseases and polygenic traits, based on emerging scientific data. In this Analysis, we model the effects of altering specific causal genomic variants on individual phenotypes and the population-wide distribution of traits. We show that the predicted effects of HPE are orders of magnitude larger than what can be achieved within one generation through embryonic selection with polygenic scores. We use these calculations to frame an ethical discussion on the implications of HPE for health equality and genetic diversity, and discuss HPE in the context of eugenics.

## Editing polygenic disease variants

We derived a mathematical model for the predicted consequences of editing specific variants linked to Alzheimer's disease (AD), schizophrenia (SCZ), type 2 diabetes (T2D), coronary artery disease (CAD) and major depressive disorder (MDD) (Methods; Supplementary Note 1 and Supplementary Tables 1–3). We used empirical estimates of the population lifetime prevalence of these common diseases, with values of 5% AD, 1% SCZ, 10% T2D, 6% CAD and 15% MDD. For illustrative purposes, we assumed that the variants identified in GWAS of these diseases are causal and investigated the effects of editing these variants on lifetime prevalence in the next generation among individuals with edited genomes. Our model differs from simulation studies by Oliynyk[20,21], who quantified the effect of "therapeutic gene therapy" on the prevalence and lifetime risk of diseases by assuming a genetic architecture and assuming a fixed change in the average polygenic score in the population (Supplementary Note 1). Our results indicate that HPE can drastically change disease prevalence among those with edited genomes. To provide an estimate of impact, we modelled the predicted reduction in prevalence among genomes that were edited at up to ten currently known GWS ($P$ value $< 5 \times 10^{-8}$) loci for each disease (Methods; Supplementary Note 1 and Supplementary Table 3). Note that the modelled results are for the proportion of the population with edited genomes and not for the entire population.

As shown in Supplementary Box 1, editing only a single variant associated with polygenic disease can have strong effects. Editing a single variant involved in AD (APOE ε4) to the protective ε2 variant is predicted to reduce the lifetime prevalence from 5%—the assumed prevalence among non-edited genomes—to 2.9% among edited genomes. This is not dissimilar to what is possible to achieve now with ESPS for APOE ε4 (which would see lifetime prevalence reduced to 3.2%). Editing ten variants (including APOE ε4) that are most strongly associated with AD, however, is predicted to reduce disease prevalence to under 0.6% (Fig. 1). Large predicted reductions in disease prevalence were also observed for SCZ, T2D, CAD and MDD. Editing ten variants with the largest effects on disease risk was predicted to reduce lifetime prevalence

from 1% to 0.1% for SCZ, from 10% to 0.2% for T2D and from 6% to 0.1% for CAD (Fig. 1). For MDD, the results are more modest; editing ten variants was predicted to reduce lifetime prevalence from 15% to 9%. The reason for the steep decrease in disease prevalence for CAD and T2D is that there are protective variants for these diseases at low frequency—edited genomes are homozygous for the protective alleles, which is in contrast to unedited genomes that are mostly homozygous for the risk alleles. The results indicate that if HPE becomes available, it would be possible to dramatically reduce the risk of common diseases in individuals with edited genomes. For example, editing 40 variants could greatly reduce an individual's lifetime risk of AD, SCZ, T2D and CAD to less than 0.2% (Supplementary Table 3). These results are far beyond what can be achieved with currently available technologies such as ESPS. Our modelling assumptions are based on the predicted effect of editing an 'average genome' in the population where prevalence, effect allelic sizes and allele frequencies are estimated. There will be variation among genomes in the predicted trait mean, before and after editing, because by chance an individual may carry more or less risk-causing alleles. This implies that not everyone benefits equally from gene editing. We quantified this by showing the standard deviations below or above the predicted changes in Fig. 1.

## Editing polygenic quantitative traits

We also considered the effect of editing multiple variants associated with quantitative traits, which are risk factors for common diseases. We identified variants associated with systolic blood pressure (SBP), diastolic blood pressure (DBP) and blood biomarkers such as fasting glucose (FG), low-density lipoprotein (LDL) cholesterol and triglycerides (TG) (Supplementary Tables 1 and 3). The predicted changes in quantitative traits were extremely large (Fig. 1). For LDL cholesterol, editing only five loci was predicted to reduce the trait value by about five phenotypic standard deviations among edited genomes, or a reduction of 2 mmol l⁻¹ (Fig. 1). The predicted effects of gene editing on LDL cholesterol and TG are very large because there are low-frequency protective variants (Supplementary Table 3). As shown in Supplementary Box 1, these cases are good candidates for gene editing intervention because nearly all unedited genomes would be homozygous for the risk variant (LDL increasing). For traits other than lipids, editing ten loci was predicted to reduce the trait by about one standard deviation among the edited genomes (for example, 10 mmHg DBP).

The results shown in Fig. 1 are for individuals with edited genomes. The predicted changes at the population level depend on the proportion of the population with edited genomes. If this proportion is small, the population-wide changes will also be small. For example, if 1% of the population has a reduction of five standard deviations in LDL due to genome editing, then the population mean is predicted to only shift by 0.05 standard deviations, or about 0.02 mmol l⁻¹.

## Model limitations and challenges

Any genetic effect is, per definition, dependent on the environment. We cannot predict future environments. However, one possible change in the context of the disease is the introduction of new treatments and therapeutics that would obviate the justification for heritable gene editing.

We quantified the effect of changing environments by modelling a gene-by-environment interaction ($G \times E$) effect, such that the genetic correlation between traits in present and future environments is less than one (Fig. 2). These results show a substantial attenuation of the predicted reduction in disease prevalence as a function $G \times E$. For some disorders, the probability of disease may depend on the prevalence in the population; therefore, it is relative and not absolute. In extreme cases, the prevalence would be constant and gene editing would not lead to any changes, consistent with genetic correlation $r_g = 0$ in the model underlying Fig. 2.

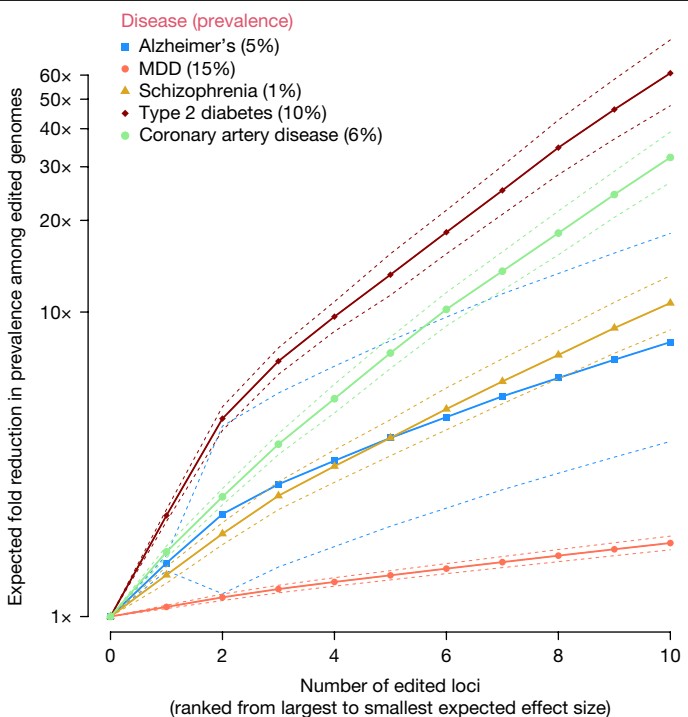

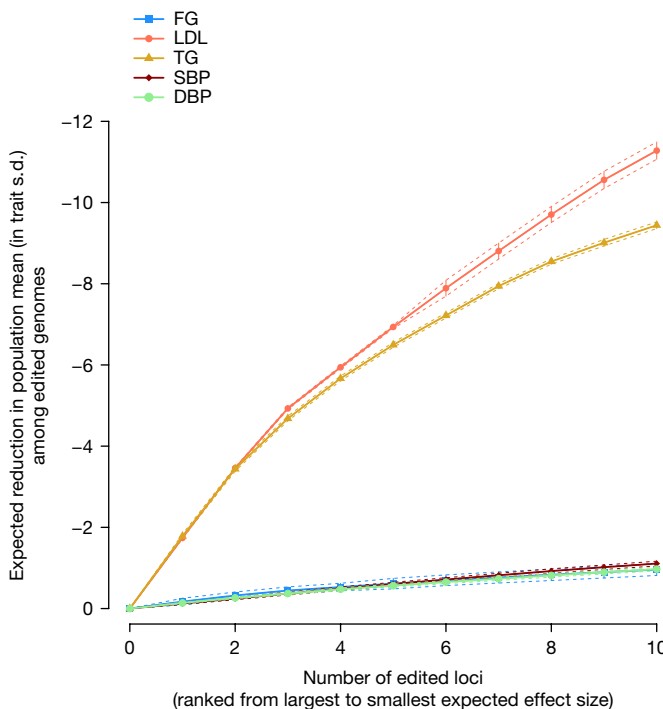

**Fig. 1 | Predicted change in phenotypic means and disease prevalence among the edited genomes.** Left, common diseases. Right, quantitative biomarkers. For each trait, a list of published GWS loci was taken. For the left panel: AD[68], MDD[69], SCZ[70], T2D[71] and CAD[72]. For the right panel: FG[73], LDL cholesterol[74], TG[74], SBP[75] and DBP[75]. The GWS loci were ordered by the product of the estimated effect size and the frequency of the undesirable allele (that is, decreasing effects for disease and biomarkers, up to a maximum of ten loci) (Methods). The x axis represents the ordered number of edited loci. The y axis represents the predicted phenotypic change among edited genomes compared to the mean of unedited genomes.

For disease, the predicted change is expressed as a fold change in lifetime prevalence. For the quantitative traits, the predicted change is in phenotypic standard deviations (s.d.). In both panels, the dotted lines correspond to standard deviations below or above the predicted changes. We calculated the predicted s.d. of gain on the liability scale as the square root of the expected variance explained by the edited loci in the general population. Expected changes of one s.d. above/below the predicted change were converted on a disease risk scale using a probit transformation. Data underlying this figure are given in Supplementary Table 1. The source code used to generate the figure is provided in 'Code availability'.

Aside from the possibility of changing environments, we made many simplified assumptions in our modelling to provide benchmarking results. Violation of these assumptions will lead to outcomes that are less than those predicted and/or lead to detrimental effects (Supplementary Notes 1 and 2).

## Simplified assumptions

First, gene editing at the scale modelled in our study is currently not feasible. Formidable barriers prevent highly multiplexed precise genome editing in eukaryotic cells at present[22]. Furthermore, gene editing technologies are known to suffer from off-target effects, which, in the context of human traits, might be considered mutagenic and, on average, likely deleterious. This places constraints on the clinical use of gene editing, especially genome editing in germ cells or embryos. We modelled the deleterious effects of off-target edits by assuming that they have a cumulative negative effect on fitness (Supplementary Note 1 and Supplementary Fig. 1). In this context, a reduction in fitness is a reduction in fertility and/or viability and an increase in mortality and morbidity. The results show that if the probability of an off-target effect is large (for example, greater than or equal to 20%) and if selection coefficients are large (greater than 0.001), then a substantial reduction in relative fitness among edited genomes is expected. However, there are reasons to take seriously the prospect of technologies that overcome these practical limitations (for example, CRISPR–Cas9 gene editing is a very active field of research[8,23]). New techniques that dramatically reduce off-target mutations are currently being developed. There are many clinical trials based on therapeutic (somatic) gene editing, and the first CRISPR therapy for sickle cell disease has been approved by the UK and US regulators[24]. Naive CRISPR–Cas9 molecules

(found in bacteria) are capable of targeting more than 200 specific genomic sequences, and engineered forms capable of making dozens of separate edits have already been produced. A future in which robust, scalable and multiplex genome editing is available is thus plausible.

Second, although the theory of gene editing is straightforward, at present, there is not a sufficiently large pool of identified causal variants for common diseases to make it feasible to apply the technology at scale. Although there are many genetic associations, they are mostly not mapped to specific functional variants. Nevertheless, the ever-increasing richness of functional genomic resources combined with new computational analysis methods and large experimental sample sizes suggests that it is reasonable to assume that many causal variants for common diseases will be identified in the next few decades. If genome-edited variants believed to be causal are not and that they have no effect on the focal phenotype, then the actual phenotypic change will be less than predicted. Supplementary Fig. 2 shows the reduction in the effect of HPE when a proportion (that is, one, two or five out of ten) of putative causal variants was misidentified and had no actual effect on the outcome when edited. For quantitative traits, the reduction in outcome compared to all ten variants being causal was proportional to the fraction misidentified. However, for disease, it can be much larger when the per-locus predicted change in liability differs among the ten loci because of the nonlinear relationship between liability and prevalence. For example, misidentifying the top 2 ranked loci for CAD changed the predicted reduction in fold change in prevalence from 32.2 to 9.5. If misidentified alleles are not causal for the focal trait but have unknown effects on one or more other phenotypes, then this would imply a risk. DNA variants that are in perfect linkage disequilibrium with causal variants, but are not causal themselves, fall into this category. If the effect sizes of polygenic traits are overestimated due to the winner's

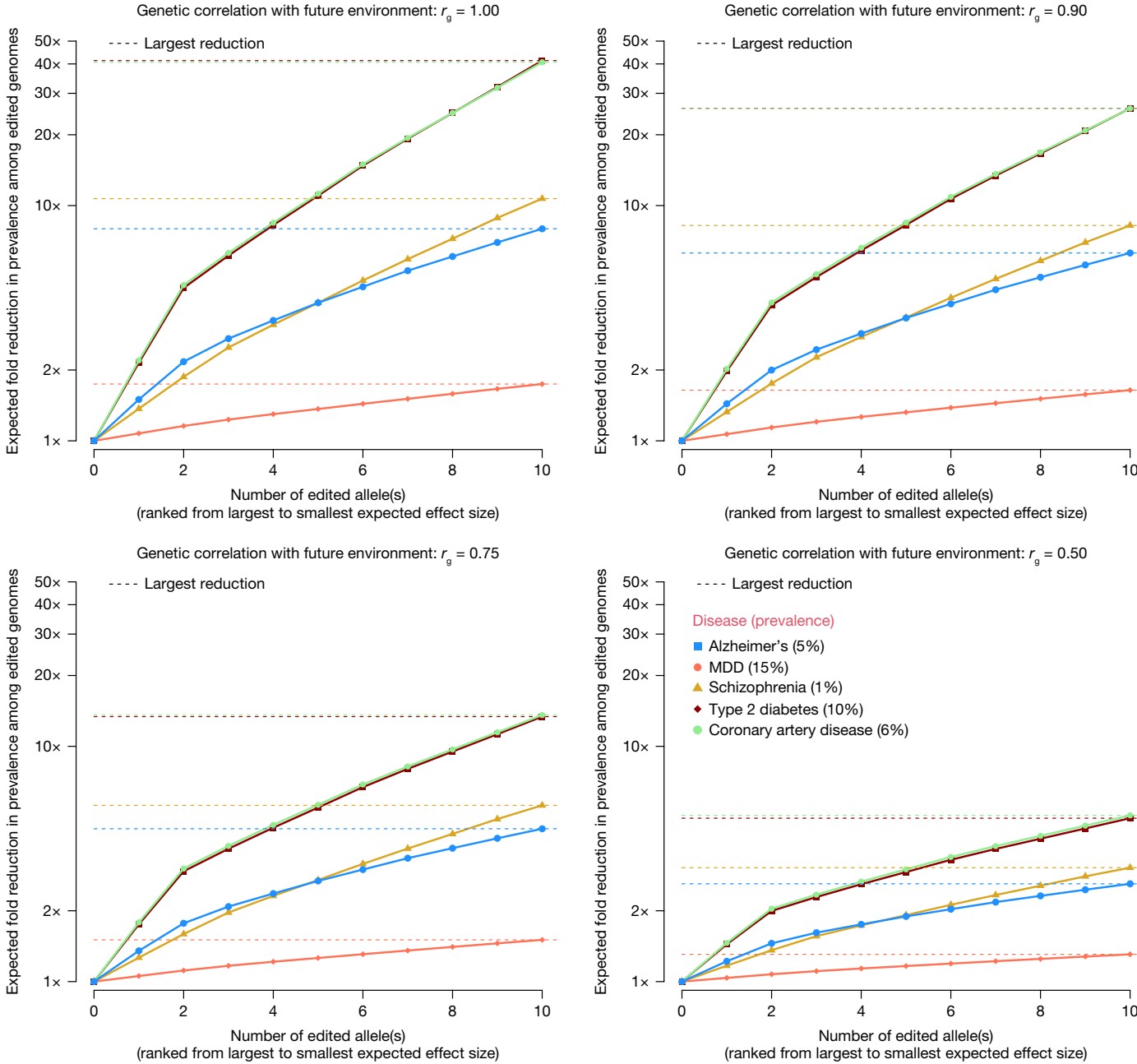

**Fig. 2 | Predicted change in phenotypic means and disease prevalence among edited genomes in the presence of gene-by-environment interactions.** We modelled $G \times E$ by allowing the $r_g$ with future environments to be reduced, such that the reduction in disease liability is shrunk by a factor of $r_g$. The source code used to generate the figure is provided in 'Code availability'.

curse (true effect sizes being smaller than estimated effect sizes due to ascertainment bias), population stratification or indirect effects, then the actual outcome of HPE will be proportionally less than predicted. For example, if effect sizes are overestimated by 10%, this will lead to a 10% reduction in outcome compared to the prediction because the latter is proportional to the effect size (Methods; Supplementary Note 1).

Third, many variants associated with polygenic diseases may have pleiotropic effects. For genomes edited for a large increase or decrease in a trait or disease, the resulting zygote may be unviable. There are known examples of variants that are protective against one disease but are risk factors for other diseases[25]. It is very difficult to prospectively predict the pleiotropic effect of new combinations of variants on prenatal development, which will be a significant source of uncertainty in the future use of HPE. Results from GWAS are consistent with the negative selection of variants associated with a wide range of traits and diseases, so that large-effect variants tend to be at lower frequency[26,27]. On an average, this predicts that inducing a large-effect change is likely deleterious. Even without a very large predicted change in any one trait, there are risks involved in focusing on a single or a combination of common disease variants. Pleiotropy is the norm in genetics. Variants associated with decreased disease can also be associated with other diseases and traits and may increase their risk. Because of pleiotropy, positive selection for one particular trait over multiple generations may lead to negative (undesirable) effects on other traits. One reason why disease risk variants are common may be that they have different roles at different times and in different cell types through development. Little is known about this kind of pleiotropy. Eliminating such risk variants may have unintended consequences on viability and fitness. We modelled a possible deleterious effect of HPE on fitness using a model of stabilizing selection (Supplementary Note 2), and the results

are shown in Supplementary Fig. 3. These results imply that the fitness of edited genomes can be substantially reduced if the phenotypic change is large and the trait is under a strong stabilizing selection. The consequences of pleiotropy mean that the actual effects of HPE on disease prevalence will be less than those predicted in this model, and the overall effects on quantitative traits not as strong as predicted.

Fourth, epistasis (gene–gene interaction) would mean that the actual effect of gene editing depends on the genomic background and, therefore, may be unpredictable. This is particularly problematic if the population in which causal variants are detected is genetically different from the population background of the edited genome. Results from GWAS in humans and from selection experiments in model organisms have shown that most genetic variation is additive by nature, which suggests that genomic background effects may not be important. However, more research is needed to quantify the effects of causal variants across genome background and environments. For example, potentially deleterious interactions between protective alleles of large effects and alleles at other loci may not be observed if they are rare. Quantitatively, the effect of epistasis would be similar to what was modelled for $G \times E$ interactions (Fig. 2), where a reduction in actual outcomes compared to what is predicted in the absence of $G \times G$ interactions.

Fifth, our modelling was based on GWAS empirical results on allelic effect sizes from naturally segregating common single nucleotide polymorphisms. These estimated effect sizes are typically small; therefore, multiple variants would need to be edited to predict a large reduction in disease prevalence (Fig. 1). Research on sickle cell disease and beta-thalassaemia has shown that large changes in fetal haemoglobin can be achieved in (somatically) gene-edited patients by targeting a transcription factor[28], even though the common allele (detected by GWAS) at that locus has a much smaller effect size[29]. This suggests that it may be possible to achieve large effects in polygenic traits among edited genomes by targeting specific regulatory elements as an alternative to targeting a causative single nucleotide of small effect found by gene mapping. Editing fewer loci might reduce risk and minimize adverse outcomes. However, the approach of targeting regulatory elements requires more biological knowledge than knowing the causative variant (for example, target gene of the variant–trait association).

Sixth, although we modelled changes among those with edited genomes, these are not directly predictive of changes in population-level disease prevalence. HPE would only be capable of drastically altering the population prevalence if a large proportion of the next generation is born through HPE. This is unlikely to occur because HPE is feasible only through in vitro fertilization. If the majority of new births continue to result from natural sexual reproduction, the potential impact of HPE on population-level disease prevalence will be small.

Seventh, somatic gene editing technologies may render heritable gene editing redundant for some diseases. Advances in somatic (therapeutic) gene editing technologies have been rapid[8,17–19], and the first CRISPR-based therapy has been approved by the UK and US regulators[24]. Somatic gene editing is an alternative to HPE and may become routine in the future for a number of diseases and risk factors, such as cholesterol[30,31]. It may also lead to risk reduction or therapy for diseases with a common allele of large effect, such as APOE ε4 and AD[32]. Somatic gene editing currently relies on biological knowledge of trait-specific genes of large effects, and it is not known whether it will progress to tackle highly polygenic diseases and traits, particularly when the relevant tissue is not known or difficult to access (for example, brain tissue). Somatic gene editing will have recurring costs for each generation, similar to other advances in treatment. Costs are currently very high, and the proposed therapies may include risks (for example, chemotherapy, if part of the treatment). It is reasonable to assume that the cost of somatic genome editing and side effects will be considerably reduced in approximately 30 years. If somatic gene editing can be performed cheaply, safely and efficiently, it may be a superior option to heritable gene editing because editing decisions could be delayed until individuals have the capacity to make their own informed decisions. On the other hand, further success and advancement of somatic gene editing may pave the way for greater acceptance and interest in heritable gene editing by demonstrating the safety and efficiency of human genetic modification. The potential of HPE to protect future generations from disease without requiring additional interventions for each generation, may be seen as an advantage that makes it a more desirable option than somatic editing.

## Ethical considerations

The prospect of HPE raises profound ethical challenges. One significant concern is that HPE will lead to renewed interest in eugenics[3]. The eugenics movement arose in Victorian Britain aiming to 'improve' the gene pool of future generations, essentially by advocating government policies that would lead to people such as those in the movement leaving more offspring[33]. This kind of eugenics has been termed 'positive eugenics'[33]. Other countries adopted 'negative eugenics' policies, which imposed severe, unethical restrictions on peoples' individual liberties (for example, forced sterilization) to prevent those considered to have 'undesirable' genes from reproducing[33]. Intellectual disability, psychiatric diseases, criminality and poverty were targets of the eugenics movement in Germany, other parts of Europe, Canada, Australia and the USA from the late 19th century to the early 20th century. Nazi eugenics was based on race and included systematic mass murder at an unprecedented scale, forced sterilization and other human rights abuses. Could HPE lead to a 21st century reincarnation of previous eugenics practices? Potentially, if it is used by non-democratic state actors, such as those that already adopt coercive control over populations.

To ensure that future uses of HPE are not eugenic, it is crucial to emphasize respect for individual liberty and societal values, such as diversity, equality and non-discrimination. A state should neither impose its vision of a good life on individuals nor use coercive measures to encourage the use of HPE. Similarly, the practice of reducing the incidence of a disease should not be equated with the notion that having a disease affects an individual's inherent moral worth. Rather, we propose that any future use of HPE should be modelled on modern clinical genetics, which uses genetic technologies to further the goals of medicine. In democratic societies, clinical genetics is voluntary and based on non-directive counselling, provision of information and choice, and interventions aimed at the well-being of the future child. When implemented with appropriate regulation and governance, HPE can be distinguished from past eugenic practices, as in contemporary clinical genetics.

Nevertheless, even if used within democratic health systems and modelled in clinical genetics, HPE may lead to undesirable outcomes for individuals and society. Since the 1970s, philosophers and bioethicists have been debating the ethical implications of altering our genetic makeup using biotechnologies[34–36]. We highlight the major ethical arguments in favour and against HPE in Table 1 from these debates[2,3,23,37–48]. The modelling we have done has direct implications for three ethical issues related to HPE (that is, 'enhancement', inequality and diversity).

## Gene editing for non-disease traits

The same techniques that enable HPE to reduce the risk of diseases can also be used to alter non-disease traits, including physical attributes, personality and cognitive traits. Using genetic technologies for purposes other than treating diseases, sometimes referred to as enhancement in the bioethics literature[49], raises specific ethical concerns. We note that human enhancement is a highly contested term, and more neutral phrasing, such as 'change of non-disease traits', may be preferable. It is sometimes assumed that to genetically change human non-disease traits, new genetic material would need to be incorporated into the genome[44]. Our modelling challenges this assumption. It indicates that HPE could lead to human phenotypes that have never been previously observed and are many standard deviations from the current mean (Fig. 1). It is conceivable, at some point in the future, that

**Table 1 | Summary of simplified ethical arguments for and against HPE**

| Ethical arguments in favour of HPE | Possible response |
|---|---|
| Consistency[2] <br> Reducing the incidence of polygenic diseases is a recognized global priority. Reducing the underlying genetic risk is similar to reducing environmental contributors to polygenic diseases. | This does not apply if the side effects of genetic interventions are different from those of environmental interventions. We lack clear evidence that reducing genetic risks has the same overall effect as reducing environmental risks. Germline gene editing introduces heritable changes; therefore, the bar for safety needs to be set higher. |
| Rights[37] <br> Future generations have the right to health. This implies a right to a low risk of polygenic diseases. | Germline genetic interventions violate the rights of future generations to make choices about their own bodies, including their genomes. |
| Distributive justice[38] <br> By reducing the incidence of polygenic diseases, HPE could reduce the strain on medical resources and make them more available to others. | This depends on the number of people who use HPE, and requires appropriately designed health systems to ensure that savings from HPE are appropriately redistributed through the health system. In addition, non-genetic interventions to promote health may better benefit the worst off. |
| Welfare[39] <br> The use of HPE can help maximize the well-being of future generations, both by lowering the risk of disease and by enhancing non-disease traits (that is, enhancement). | Using HPE to increase individual well-being through a market exacerbates injustice and inequality and reduces valuable forms of diversity. Currently, genetic interventions are extremely expensive. Changes in the environment and non-genetic interventions may be cheaper and more accessible. |

| Ethical arguments against HPE | Possible response |
|---|---|
| Inequality[3] <br> HPE could deepen inequalities in future societies. Disease risk could be concentrated in those with lower socio-economic groups who are already most disadvantaged. | This depends on how HPE is regulated and made available. There are ways to implement HPE, where it reduces existing inequalities by prioritizing the worst off, as should be done with all medical treatments (for example, through public funding). |
| Safety <br> HPE introduces new combinations of variants that can be dangerous or unsafe. It would be unethical to impose this uncertain risk on future generations. | It is vital that any use of HPE be supported by rigorous safety data and have a clear justification[23] through a risk/benefit balance. In addition, natural reproduction generates new combinations of variants. One strategy is to limit HPE to variant combinations already seen in existing populations. |
| Enhancement[40] <br> HPE can be used to select non-disease traits (for example, intelligence or athletic ability) and produce human phenotypes that we have never seen before. This may cause future generations to be very different from the current generation. | HPE for non-disease traits can be potentially controlled through regulations, which limit its purpose to reducing risk of disease and promoting well-being. |
| Diversity[41] <br> Widespread use of HPE can reduce the valuable forms of diversity. | Targeting polygenic traits may have a limited effect on genetic diversity. The value of genetic diversity should be weighed against other values, such as individual well-being and health. |
| Means matter[42] <br> HPE may cause us to overlook other approaches for reducing polygenic diseases. For example, preventing heart disease through HPE may reduce efforts to improve diet and exercise. | This is not an either/or option. Society can prioritize environmental measures that are low cost and broadly beneficial while also lowering genetic risk. |
| Design[43] <br> Human polygenic traits have been carefully designed through natural selection. The use of HPE to alter human bodies disrupts this design and leads to harm. | Human bodies have mutations that predispose them to diseases and disadvantages. Evolution is blind to human suffering. The fact that so many people suffer from polygenic diseases shows the need to intervene and reduce suffering. |
| Non-identity[44] <br> Large-scale polygenic editing may be 'identity-altering'. The non-identity problem is a puzzle in ethics when our current actions might change who is born in the future. It can be hard to specify how these actions can be harmful to people who would otherwise not have existed. This means that it does not benefit individuals but rather changes who exists. | Many public health interventions may also be identity-altering. An example is delaying conception to avoid Zika infection[45], which changes the timing of conception and, thus, the identity of the sperm and egg creating a child. However, this does not mean that we should not consider the well-being of those who come into existence, even if the alternative is non-existence. At most, this indicates that we should give lower priority to HPE than to identity-preserving medical interventions (for example, antibiotics and surgery). |
| Expressivist[46] <br> The use of HPE to eliminate variants linked to a disease or disadvantage expresses a negative attitude towards individuals living with that disease or disadvantage. | HPE does not need to express a negative attitude towards people, but rather towards diseases or traits. We should ensure that support is maintained for those who do not receive gene editing. |
| Reduced resources[47] <br> The use of HPE to eliminate variants linked to a disease or disadvantage may reduce support for others living with conditions, who cannot or do not wish to obtain access to HPE. | This is not an inevitable consequence of HPE. In some cases, reducing the incidence of a disease will result in more resources available to those who remain with conditions. For example, a reduction in the incidence of beta-thalassaemia in Cyprus due to carrier screening programmes resulted in more resources available to the remaining patients[48]. |

HPE could be used to target traits, such as height and intelligence, and lead to large-scale changes in these traits. Although human populations have undergone dramatic changes over the past few generations as a result of cultural and environmental changes, the prospect of radical, rapid changes in human physiology raises unique ethical concerns[50]. Future populations with radically different physiologies and psychologies may develop very different values from those living today. Human change in non-disease traits could change society in unprecedented ways and not necessarily for the better. Furthermore, using HPE to make individuals 'better than well'[51] can be seen as unfair in a world where many people do not have access to adequate healthcare.

It is currently possible to test embryos created through in vitro fertilization for their predisposition to non-disease traits. However, many jurisdictions only allow embryos to undergo genetic testing and screening to prevent a serious disease[52]. Nevertheless, when considering polygenic traits, the line between health and disease is blurred[5]. For example, is using HPE to reduce blood pressure, a causal risk factor for common diseases, a medical or non-medical application? The same question arises regarding vaccines and other preventative interventions. In extreme cases, polygenic editing can be used to delay normal human ageing, significantly prolonging human life.

One approach could be to limit the use of HPE to cases in which there is a reliable relationship between a trait and positive effects on well-being[53]. Of course, this raises the vexed issue of which conception of well-being to use, but this is a problem for any welfarist approach to individual or societal improvement.

Another possibility is to limit HPE to combinations of protective alleles that naturally occur in today's populations. Some people alive

today possess great genetic resistance to polygenic diseases. For example, the chance of an individual carrying ten protective alleles against AD (thus having a risk of 0.3%) is one per two billion, indicating that there may be people alive today with this combination. Similarly, there may be people alive today who enjoy genetic protection against a wide range of polygenic diseases. These already existing combinations of protective variants could be the targets of HPE. In such cases, the goal is to provide protection to those in a population with the highest risk of developing a polygenic disease, similar to those with the lowest genetic risk. This would be egalitarian and promote genetic equity, where HPE could be used to make people as healthy as the healthiest people living today.

Data on public attitudes towards genome editing suggest other concerns. While most countries are strongly against using gene editing technology to increase intelligence, others are not. A recent study analysing public attitudes towards genome editing around the world found one country outlier regarding support for targeting non-disease traits. In India, 64% of respondents were in favour of germline editing and intelligence—far higher than in any other country surveyed[54]. A recent survey of 6,800 people in the USA reported that about 40% of respondents deemed heritable gene editing for medical and non-medical traits either 'not a moral issue' or 'morally acceptable'[55], and only 17% believed it was wrong. If HPE becomes a possibility, perhaps we should consider a future where HPE is restricted in some countries but unrestricted in others. Countries might feel pressure to allow HPE for non-disease traits out of fear of being outcompeted by countries that embrace the technology.

## Inequality

An enduring concern regarding genetic and other technologies is that they will increase inequalities, making the dominant social class the dominant biological class, as depicted in the film *Gattaca*[56]. Although many technologies are initially only accessible to those that can afford them, there is reason to be concerned about differential access to HPE. Inequalities in wealth are substantially social in nature, reflecting unequal access to resources and opportunities, which can be corrected. HPE could write these inequalities into our biology.

Our modelling gives substance to these concerns by showing that dramatic changes can be achieved through HPE. Individuals with edited genomes may have a much lower risk of disease than those with unedited genomes. If HPE is only available to those in higher socioeconomic groups, then this will more heavily skew the disease burden of polygenic diseases to those who are already the worst off[57]. Diseases such as depression and heart disease may become diseases that only occur in certain demographics.

The unequal use of HPE is likely to increase social division. In Fig. 3, we quantify the increase in inequality of the risk of diseases as a function of the proportion of the population undergoing HPE. Additional results for several hypothetical diseases are also given in Supplementary Fig. 4. These results show that there is an increase in inequality, as measured by the Gini index, when a small proportion of the population carries edited genomes and that inequality is only decreased when more than 50% of the population has edited genomes.

Existing genetic variations contribute to social and health inequalities. For example, people with more alleles associated with higher educational attainment are more likely to migrate to locations with better economic opportunities. This leads to an increase in social stratification[58], which, when combined with the common practice of assortative mating on traits associated with educational attainment, can increase the phenotypic variation in the population by as much as 20% (ref. 59), thereby increasing health inequalities for common diseases genetically correlated with educational attainment. Reducing health inequalities caused by a random genetic lottery[60] seems fair and desirable. Providing equitable access to new technologies, such as GWAS and their downstream clinical translations today[61] and, perhaps, HPE in the future, could reduce health inequalities.

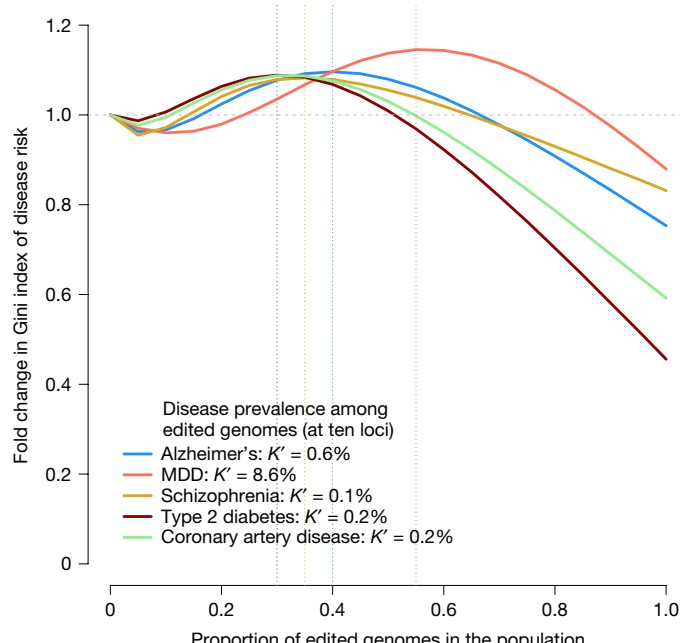

**Fig. 3 | Quantification of health inequality in a population that includes a fraction of genome-edited individuals.** We modelled the probability of disease in the population as a mixture distribution with two components: one component with a reduced risk, representing the fraction of edited genomes in the population, and another component representing unedited genomes (Methods). Diseases and prevalence among non-edited genomes are the same as those shown in Fig. 1. The prevalence among edited genomes ($K'$) was taken from Fig. 1, assuming ten edited loci. The $x$ axis represents the fraction of edited genomes in the population, varying from 0 to 1, and the $y$ axis represents the relative Gini index in the population compared to a population with no edited genomes. The vertical dotted lines indicate which fraction of edited genomes in the population yields the maximum Gini index for each disease (that is, maximum risk inequality). The source code used to generate the figure is provided in 'Code availability'.

Reducing the incidence of polygenic diseases could further lead to a more equitable distribution of health resources within health systems. Common chronic diseases, such as heart disease and psychiatric disorders, are the main contributors to the global cost of healthcare and contribute substantially to the loss of disability-adjusted life years. Reducing the amount of health resources spent on fighting these diseases could free up resources that could be reinvested in other health priorities. Reducing the number of people whose health depends on access to resources will free up resources for others in need.

The implications of international inequality may be more difficult to control. However, many countries are unlikely to have the capacity to use HPE. If high-income countries use HPE, this could result in polygenic diseases becoming even more concentrated in developing countries.

There are no easy solutions to these problems, which is why it is vital that we start to consider the implications now while the prospect of HPE is still many years away.

## Diversity

One concern in human genetic engineering is that this will lead to a loss of genetic diversity[62,63]. Examples of the dangers of a lack of genetic diversity are found in agriculture. Modern crops have been selected for their enhanced efficiency in food production but tend to be more susceptible to disease epidemics and have a reduced capacity to adapt to changes in environmental conditions. Similar concerns have been expressed regarding the use of heritable gene editing in humans. In the pursuit of healthy, happy children, there is fear that we might create genetically homogeneous human populations with increased

susceptibility to disease and decreased potential for adaptation to future risks[63]. However, our modelling suggests that these concerns are mostly unfounded for HPE. For the traits considered, there was a high degree of background genetic diversity, with tens of thousands of loci responsible for the observed genetic variation. Relatively few genetic changes are needed to make very large changes to phenotypes and reduce disease risks. This suggests that it is possible to radically reduce the risk of polygenic diseases in human populations while maintaining high levels of genetic diversity. Moreover, modern approaches to disease prevention and treatment, such as the human immunodeficiency virus and Covid-19 pandemics, do not rely on genetic diversity and resilience, but on the application of science to develop therapeutic and biopsycho-social interventions to manage disease. Indeed, HPE or somatic gene editing can, in theory, be applied to confer genetic resistance.

**A collectivist ethical approach.** The use of genome editing technologies by individuals and couples will affect individuals' genomes, which will affect their whole lives in multiple ways, as well as the human gene pool. This will require a holistic evaluation of the effect on a whole life that requires a benchmark of a good human life, which is a topic of thousands of years of philosophical debate. However, this also suggests that there may be limitations to ethical approaches to HPE that focus solely on its effect on individuals.

One alternative ethical perspective is based on collective welfarism. According to this approach, the goal of biotechnology should be to provide benefits to individuals and to broader groups of individuals, including families, communities and societies[4]. From this perspective, it is also important that HPE not be implemented in ways that decrease social cohesion, increase division and weaken our communities and society. Notably, this approach requires further analysis of what constitutes flourishing societies.

In the long term, there may be an obligation to pursue and develop technologies such as HPE. Mildly deleterious mutations that escape natural selection because of better medical care are predicted to accumulate in the gene pool[64]. Previously published models suggest that the effect of this 'genetic load' might manifest itself as physical and mental deterioration in only a few generations[64]. However, this concept is controversial, and the conclusions are debated[65–67]. If we take seriously the idea of leaving future generations in a better state than the current generations, then we have reason to provide them with the preconditions for a good life. This includes access to clean water, unpolluted air, education and shelter, and may include the use of HPE to lower the genetic risk of disease.

Although collectivist considerations should inform the values of governments and the goals they pursue, it is also important that these goals do not override basic human rights, such as the right to autonomy. The pursuit of collectivist goals must be compatible with basic human rights.

## Pathway to polygenic editing

Before any human use of heritable gene editing is considered, it is necessary to first show that it is safe and effective in animal models, particularly non-human primates. An international commission from The National Academies of Sciences, Engineering, and Medicine produced guidelines in a study report 'Heritable Human Genome Editing'[7]. The commission's pathway takes lack of clinical alternatives to be the most important criterion when selecting initial targets for human heritable gene editing, whereas we propose 'expected benefit'. The commission's category A (initial targets for human heritable gene editing) includes editing to prevent adult-onset diseases, such as Huntington's disease, and treatable conditions such as cystic fibrosis (in cases where preimplantation genetic diagnosis is not feasible). In contrast, we propose that these conditions should only be targeted after successful gene editing for lethal, untreatable, childhood-onset disease[49].

We propose that the first human use of heritable genome editing could be performed in single-gene disorders that are lethal early in life (for example, BRAT-1 and Tay–Sach's Disease). In such cases, gene editing could be life-saving, and there is low expected harm (probability of harm × magnitude of harm) and great expected benefit (probability of benefit × magnitude of benefit)[49]. Safety can be assessed by long-term follow-up, embryo biopsy and prenatal testing, including whole genome sequencing and morphological assessment. After lethal single-gene disorders, it can be extended to severe childhood-onset single-gene disorders, such as cystic fibrosis or thalassaemia major, and if shown to be safe, it can be extended to incurable and unpreventable adult-onset disorders (for example, Huntington's disease). If shown to be safe and effective in these early-onset or incurable and unpreventable disorders, it could be extended to severe adult-onset disorders where some screening, prevention and treatment exist, although with significant morbidity (for example, BRCA breast cancer and familial adenomatosis polyposis). Before such applications are considered, it is important to conduct further research on the effect of polygenic variants on individuals in natural populations, including the lifelong consequences of carrying rare protective alleles. The next stage would involve limited polygenic editing, for example, a small number of variants contributing to AD. The number of genes edited could be increased as the safety and effectiveness profiles emerge at each stage.

## Concluding remarks

Advances in technology[23] have already led to the birth of at least two genetically edited children and children screened (before birth) for polygenic conditions. Over the coming decades, it may become possible to make multiple edits to the DNA sequence of human embryos and germ cells, potentially targeting dozens to hundreds of variants involved in the development of complex traits. In this Analysis, we demonstrated that editing a relatively small number of variants could make dramatic changes to an individual's risk of disease, and if widely and safely used, it may substantially reduce the incidence of polygenic diseases among those with edited genomes.

From the modelling results, it appears that editing only a few variants would maximize benefit and minimize risk, so that an 'oligogenic' approach may be preferred to an approach with many loci. However, there are still too many unknowns to draw such a strong conclusion. For example, we do not know how gene editing technologies will develop with respect to precision (that is, risk of deleterious off-target effects) in the next 30 years. There may be diseases and their risk factors that do not have rare protective variants with large effects, so that change may only be achieved by editing many loci. Rare large-effect variants may also show deleterious epistatic effects when homozygous.

Gene editing techniques applied to non-disease traits may deepen inequalities and raise the spectre of eugenics. It is vital for governments and the international community to carefully consider how to regulate HPE to best manage the ethical challenges. In doing so, it is important to consider the risk of deciding not to use HPE. Polygenic diseases are a leading cause of premature death worldwide, strain the health system and reduce people's freedom by making them reliant on medical resources. Successful management of the risks posed by HPE will likely require strong international cooperation, which is particularly challenging in the face of globally competing interests, priorities and conflicting values. There is good reason to start exploring the challenges and opportunities that HPE provides now, well before it becomes a practical possibility, and our modelling serves as a foundation for an informed and balanced discussion on the potential use of gene editing to reduce the genomic contribution to common diseases or traits.

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

## Methods

A summary of the methods is described below. More details are given in Supplementary Note 1.

### Calculations

We assume $m$ causal variants with additivity within and between loci and genotypes in Hardy–Weinberg and linkage equilibrium. We define trait-increasing allele frequency as $p_i$ ($i = 1 \dots m$) and trait-increasing effect size as $\beta_i$ (that is, $\beta_i > 0$). If $x_i$ is the number of trait-increasing alleles ($x_i = 0$, 1 or 2) at causal (or trait-associated) single-nucleotide polymorphism (SNP) $i$, then the expectation and variance (under Hardy–Weinberg equilibrium) of $x_i$ are $E[x_i] = 2p_i$ and $var[x_i] = 2p_i(1 - p_i)$.

The mean of phenotype $Y$ in the population can be expressed as $E[Y] = \mu + \sum_{i=1}^{m} 2p_i\beta_i$, where $\mu$ is a constant. If germline gene editing were to be applied to all $m$ loci, by making all loci homozygous for either the trait-increasing alleles (that is, all $x_i = 2$ and $p_i = 1$ among edited genomes) or the trait-decreasing allele (that is, all $x_i = 0$ and $p_i = 0$ among edited genomes), then the expected phenotype of a gene-edited genome would be $\mu + \sum_{i=1}^{m} 2\beta_i$ for all trait-increasing alleles and $\mu$ for all trait-decreasing alleles. Hence, the difference in phenotypic means between the current population and the one after gene editing is $\sum_{i=1}^{m} 2(1 - p_i)\beta_i$ for homozygosity of trait-increasing alleles and $-\sum_{i=1}^{m} 2p_i\beta_i$ for homozygosity of trait-decreasing alleles. These expressions were used to predict mean phenotype changes for the quantitative traits (Fig. 1), using results from GWAS (below).

To model the effect of gene editing on a disease or disorder, we assume a liability threshold model. Liability ($\ell$) for multi-locus genotype $g$ is defined as $\ell = g + e$, where $E[\ell] = E[g] = E[e] = 0$ and $var[\ell] = 1$. We denote the lifetime prevalence of disease in the current population as $K$. The probability ($P$) of disease ($D$) given genotype $x$ can be expressed as $P(D,|,x) = 1 - \Phi(t - \mu_x)$, where $\mu_x$ denotes the average liability of individuals with that particular genotype, $\Phi$ is the cumulative distribution function of standard Gaussian distribution and $t = \Phi^{-1}(1 - K)$ is the threshold corresponding to lifetime prevalence $K$.

For a single locus, $P(D,|,x) = 1 - \Phi[t - (x - 2p)\beta]$, where $x = 0$, 1 or 2, and $\beta$ is the effect in standard deviation units on the liability scale of the risk-increasing variant. After gene editing to reduce disease risk, $x = 0$ for the target variant and $K_g = P(D,|,x = 0) = 1 - \Phi[t + 2p\beta]$ is the prevalence of disease among edited genomes. For $m$ edited loci, $K_g = 1 - \phi\left[t + \sum_{i=1}^{m} 2p_i\beta_i\right]$, which is the equation used to generate results for the disorders in Fig. 1.

### Data from GWAS

We used lists of GWS loci for multiple disorders and risk factors (Supplementary Tables 1–3). GWS loci for AD[68], MDD[69], SCZ[70], T2D[71] and CAD[72] were collected, and for the quantitative traits we considered FG[73], LDL cholesterol[74], TG[74], SBP[75] and DBP[75]. The effect sizes for disorders were reported in the natural logarithm of the odd ratio units, $\beta_{\log(OR)}$, then transformed to a scale of liability using $\beta = \beta_{\log(OR)}K(1 - K)/z$, where $K$ is the lifetime prevalence, and $z = \varphi(t)$ is the density of a standard normal distribution calculated for $t = \phi^{-1}(1 - K)$.

For each trait and disorder, the GWS loci were ordered by the product of their estimated effect size and the frequency of the risk-decreasing allele, up to a maximum of ten loci (Supplementary Table 3). Allele frequencies were taken from the published papers. For quantitative traits, we estimated the phenotypic variance from the reported sample sizes and standard errors (s.e.) using the mean of $2p_i(1 - p_i)N_i[\text{s.e.}_i^2]$ across all reported GWS loci, where $p_i$, $N_i$ and s.e._$i$ are the reported allele frequency, sample size and s.e. for locus $i$. We then expressed the estimated effect sizes in phenotypic standard deviation units. Estimating the phenotypic s.d. was necessary because not all GWAS papers used standardized trait values. The extracted data from the GWAS papers can be found in Supplementary Tables 1 and 2 and in the 'Code availability' for Fig. 1.

### Effect of $G \times E$ on predicted disease prevalence among edited genomes

We modelled the effect of G × E by allowing the genetic correlation ($r_g$) across current and future environments to be less than 1.

For $m$ edited loci and given $r_g$, $K_g(r_g) = 1 - \Phi\left[t + r_g \sum_{i=1}^{m} 2p_i\beta_i\right]$, which is the equation used to generate Fig. 2. The same equation that is relevant when editing genomes results in gene–gene interactions (epistasis) such that the genetic correlation between edited and unedited genomes is equal to $r_g$.

### Effect of genome editing on health inequality in the population

We modelled the risk of disease in the population using a liability threshold model. As before, $K$ is the disease prevalence among unedited genomes. We calculated the risk of disease $R_i$ for each individual $i$ in the population as $R_i = 1 - \Phi(\ell_i + \mu_E E_i)$, where $\ell_i$ is the (unobserved) disease liability of individual $i$, $E_i$ is an indicator variable equal to 1 if individual $i$ genomes have been edited and 0 otherwise, and $\mu_E$ is the mean liability among edited genomes. We simulated a population with $n = 1{,}000{,}000$ individuals and varied the fraction of edited genomes from 0 to 1. We then calculated the Gini index of the resulting distribution of the probability of disease under various scenarios corresponding to different fractions of edited genomes, different disease prevalence and different objectives of editing, including a reduction of disease prevalence by 10-, 100- or 1,000-fold. Gini indexes were calculated in R (v.4.3.0) using the DescTools package.

### Reporting summary

Further information on research design is available in the Nature Portfolio Reporting Summary linked to this article.

## Data availability

Data used to generate Figs. 1–3 presented in this paper are available at GitHub (https://github.com/loic-yengo/Code_for_GeneEditing_Paper_Visscher_et_al_2024) and Zenodo (https://doi.org/10.5281/zenodo.7513325)[76].

## Code availability

Publicly available software tools were used for all analyses. These software tools are listed in the main text and in Methods. R scripts to generate Figs. 1–3 presented in this paper are available at GitHub (https://github.com/loic-yengo/Code_for_GeneEditing_Paper_Visscher_et_al_2024) and Zenodo (https://doi.org/10.5281/zenodo.7513325)[76].

76. Visscher, P., Gyngell, C., Yengo, L. & Savulescu, J. Heritable polygenic gene editing: the next frontier in genomic medicine? *Zenodo* https://doi.org/10.5281/zenodo.7513325 (2024).

**Acknowledgements** P.M.V. and L.Y. are supported by the Australian Research Council (FL180100072, DE200100425 and FT220100069). J.S. is supported by grants from the Wellcome Trust (grant number 226801) and Australian Research Council (LP190100841). For the purpose of open access, the author has applied a CC BY public copyright licence to any Author Accepted Manuscript version arising from this submission. Through their involvement with the Murdoch Children's Research Institute, J.S. and C.G. receive funding from the Victorian State Government through the Operational Infrastructure Support programme. J.S. and C.G. are also supported by the Australian Government through the Medical Research Future Fund as part of the Genomics Health Futures Mission (grant number 76749). The views expressed herein are our own and are not necessarily those of the funding bodies. We thank N. Wray, D. Benjamin and P. Turley for thoughtful discussions and many helpful comments and suggestions on an earlier version of the paper.

**Author contributions** P.M.V. and J.S. conceived the idea of writing a paper on this topic. P.M.V. and L.Y. performed theoretical calculations. L.Y. conceived ideas for generating figures and

created all figures. All authors contributed to writing multiple revisions of the paper and writing detailed responses to the reviewers' comments.

**Competing interests** J.S. is a Partner Investigator on an Australian Research Council grant LP190100841, which involves an industry partnership from Illumina. He does not personally receive any funds from Illumina. He is a Bioethics Committee consultant for Bayer and a Bioethics Advisor to the Hevolution Foundation. The other authors declare no competing interests.

**Additional information**

**Correspondence and requests for materials** should be addressed to Peter M. Visscher or Julian Savulescu.

# Reporting Summary

## Statistics

For all statistical analyses, confirm that the following items are present in the figure legend, table legend, main text, or Methods section.

| n/a | Confirmed | |
|---|---|---|
| ☐ | ☒ | The exact sample size ($n$) for each experimental group/condition, given as a discrete number and unit of measurement |
| ☒ | ☐ | A statement on whether measurements were taken from distinct samples or whether the same sample was measured repeatedly |
| ☒ | ☐ | The statistical test(s) used AND whether they are one- or two-sided *Only common tests should be described solely by name; describe more complex techniques in the Methods section.* |
| ☒ | ☐ | A description of all covariates tested |
| ☐ | ☒ | A description of any assumptions or corrections, such as tests of normality and adjustment for multiple comparisons |
| ☐ | ☒ | A full description of the statistical parameters including central tendency (e.g. means) or other basic estimates (e.g. regression coefficient) AND variation (e.g. standard deviation) or associated estimates of uncertainty (e.g. confidence intervals) |
| ☒ | ☐ | For null hypothesis testing, the test statistic (e.g. $F$, $t$, $r$) with confidence intervals, effect sizes, degrees of freedom and $P$ value noted *Give P values as exact values whenever suitable.* |
| ☒ | ☐ | For Bayesian analysis, information on the choice of priors and Markov chain Monte Carlo settings |
| ☒ | ☐ | For hierarchical and complex designs, identification of the appropriate level for tests and full reporting of outcomes |
| ☒ | ☐ | Estimates of effect sizes (e.g. Cohen's $d$, Pearson's $r$), indicating how they were calculated |

*Our web collection on statistics for biologists contains articles on many of the points above.*

## Software and code

Policy information about availability of computer code

| | |
|---|---|
| Data collection | GWAS summary statistics were downloaded from the papers cited in the manuscript. |
| Data analysis | Data and R scripts underlying all Figures are available on GitHub (https://github.com/loic-yengo/Code_for_GeneEditing_Paper_Visscher_et_al_2024). Allele frequencies of SNPs significantly associated with Major Depression Disorder, Alzheimer's Disease and Schizophrenia used in our study were calculated using PLINK v1.9 from a set of 348,501 unrelated European ancestry participants of the UK Biobank (as previously described - Yengo et al. HMG (2018)). Analyses were performed using R version 4.2.0. |

For manuscripts utilizing custom algorithms or software that are central to the research but not yet described in published literature, software must be made available to editors and reviewers. We strongly encourage code deposition in a community repository (e.g. GitHub). See the Nature Portfolio guidelines for submitting code & software for further information.

## Data

Policy information about availability of data

All manuscripts must include a data availability statement. This statement should provide the following information, where applicable:
- Accession codes, unique identifiers, or web links for publicly available datasets
- A description of any restrictions on data availability
- For clinical datasets or third party data, please ensure that the statement adheres to our policy

All GWAS summary statistics used in the paper are available in the public domain. Data (i.e., specific sets of SNP, effect sizes and frequencies) and R scripts

underlying all Figures are publicly available on GitHub (https://github.com/loic-yengo/Code_for_GeneEditing_Paper_Visscher_et_al_2024). UK Biobank data were accessed under application 12505.

## Human research participants

Policy information about studies involving human research participants and Sex and Gender in Research.

| | |
|---|---|
| Reporting on sex and gender | N/A |
| Population characteristics | Allele frequencies of SNPs significantly associated with Major Depression Disorder, Alzheimer's Disease and Schizophrenia used in our study were calculated using PLINK v.19 from a set of 348,501 unrelated European ancestry participants of the UK Biobank (as previously described - Yengo et al. HMG (2018)). |
| Recruitment | UK Biobank investigators sent postal invitations to 9,238,453 individuals registered with the UK's National Health Service who were aged 40–69 years and lived within approximately 25 miles (40 km) of one of 22 assessment centers located throughout England, Wales, and Scotland. Overall, 503,317 participants consented to join the study cohort and visited an assessment center between 2006 and 2010, resulting in a participation rate of 5.45%. (Fry et al. Am J Epidemiol. 2017 Nov 1;186(9):1026-1034). |
| Ethics oversight | The National Information Governance Board for Health and Social Care and the North West Multicentre Research Ethics Committee provided approval for UK Biobank to obtain the contact details of people within the eligible age range from local National Health Service Primary Care Trusts. |

Note that full information on the approval of the study protocol must also be provided in the manuscript.

## Field-specific reporting

Please select the one below that is the best fit for your research. If you are not sure, read the appropriate sections before making your selection.

☒ Life sciences          ☐ Behavioural & social sciences          ☐ Ecological, evolutionary & environmental sciences

For a reference copy of the document with all sections, see nature.com/documents/nr-reporting-summary-flat.pdf

## Life sciences study design

All studies must disclose on these points even when the disclosure is negative.

| | |
|---|---|
| Sample size | Allele frequencies of SNPs significantly associated with Major Depression Disorder, Alzheimer's Disease and Schizophrenia used in our study were calculated using PLINK v.19 from a set of 348,501 unrelated European ancestry participants of the UK Biobank (as previously described - Yengo et al. HMG (2018)). |
| Data exclusions | We used two exclusion criteria for calculating allele frequencies: (1) Non-European ancestry individuals and (2) related individuals. Ancestry and relatedness was inferred using called genotypes at imputed SNPs as previously described in Yengo et al. (2018). |
| Replication | N/A |
| Randomization | N/A |
| Blinding | N/A |

## Reporting for specific materials, systems and methods

We require information from authors about some types of materials, experimental systems and methods used in many studies. Here, indicate whether each material, system or method listed is relevant to your study. If you are not sure if a list item applies to your research, read the appropriate section before selecting a response.

## Materials & experimental systems

| n/a | Involved in the study |
|-----|----------------------|
| ☒ | Antibodies |
| ☒ | Eukaryotic cell lines |
| ☒ | Palaeontology and archaeology |
| ☒ | Animals and other organisms |
| ☒ | Clinical data |
| ☒ | Dual use research of concern |

## Methods

| n/a | Involved in the study |
|-----|----------------------|
| ☒ | ChIP-seq |
| ☒ | Flow cytometry |
| ☒ | MRI-based neuroimaging |

