## [Peer Review File · Nature]

Heritable polygenic editing: the next frontier in genomic medicine?

Corresponding Author: Professor Peter Visscher

Version 1:

Reviewer comments:

Referee #1

(Remarks to the Author)

As someone whose daily practice for the past 2 decades has been building genome editing as a tool for clinical intervention, this reviewer was prepared to be concerned about the submitted manuscript as distracting from the immediate cause of expanding the public health impact of CRISPR on existing disease. To the contrary, the manuscript is a thoughtful and cogent extended argument in support of the authors' statement (line 80) that "it is vital to discuss the practical, ethical and social implications of PGE now." The work is timely and of broad interest, and the authors are to be commended for taking on such a thorny subject.

From the standpoint of this reviewer's area of immediate technical expertise the major issue – that can be addressed – of the submitted manuscript is a certain level of separation from the actual clinical practice of genome editing today. Suggestions for manuscript improvement to that effect are below. Separately, a significant concern – detailed below – is the authors' discussion of PGE for IQ enhancement. In the eye of this reviewer, this passage requires significant revision.

61 and elsewhere: as the authors are doubtless aware, the CRISPR-based approach that is closest to approval as a prescribable medicine is editing a GWAS hit, BCL11A (30355263). Of direct relevance to their argument is the fact that the natural allele of BCL11A is weak and changes HbF by ~0.5 g/dl (out of a total of 15 g/dl) – ref 18245381. The edited allele – as measured in gene-edited patients – changes HbF by 7 d/gl (33283989). This proves that editing can make a stronger allele of a GWAS hit than occurs in Nature. It is highly likely that such "BCL11A enhancer"-like targets would be found for common disease (as is the case for CAD and PCSK9 or ANGPTL3).

67-78 and 180-181 and 193: the authors could consider citing 32315033 and 36064968 as an example of multilocus editing in tissue culture cells that has already been attained. The other example to consider citing is the work from Beam 35560156 – it's a clinically relevant cell and this approach is in the clinic. The "multieditable human genome" future the authors allude to is quite close.

93 and elsewhere: the authors should consider mentioning somatic genome editing as a clear alternative. Eg it is important to discuss the ongoing clinical trial by Verve to knock out PCSK9 in the liver (preclinical data 34012082) for familial CAD. Verve has shown strong preclinical data that they can double-edit PCSK9 and ANPTL3 in nonhuman primates. This means that in very practical terms, there will be methods to deliver a CAD-protective genome edit during the postnatal period – without any of the handwringing about embryo PGE. A gene therapy trial has begun to deliver ApoE2 to patients homozygous for ApoE4 (29409358) – and this will assuredly pave the way for editing ApoE4 directly (eg <https://www.businessinsider.com/gene-editing-pioneer-david-liu-developing-drug-to-prevent-alzheimers-2021-11>). It is essential the authors make it clear that such approaches – especially for diseases that are later-onset (eg CAD or AD) – can be ethically advanced through clinical development today, in full compliance with existing laws for protection of human subjects in research, the Belmont report, etc. Thus, by the time – 30 years from now in the authors' estimate – that embryo editing will technically get to the point to allow safe PGE, there will be a number of approved medicines to reduce risk of common diseases by editing adult individuals who have given informed consent.

442: further to the point just made, it would be most appropriate to re-cite the recent Doudna review here. CRISPR has been marching through clinical development with trials for sickle, thal, TTR, HAE, LCA, HIV, CAD, cancer – and a number of others to initiate shortly. This is not tangential to the future the authors describe – it's enabling. The only path to PGE for, say,

CAD, is an extended track record for such editing in the postnatal period to treat established disease. When and only when such clinical use is shown to be safe and effective will there come a time to discuss PGE.

134: the authors' discussion of using PGE to increase IQ is – in its present form – gravely lacking in key details.

The paper the authors cite identifies 205 genomic loci bearing 1,016 genes as exerting an effect on “IQ” (however one defines that – perhaps “reifies” is a better term) in individuals of European ancestry. The notion that a trait of such complexity can be manipulated by using editing to “introgress 20 smarter alleles” into the genomes of the “unintelligent” is simply untenable. In contrast to diseases such as CAD or neurodegeneration, there is, by definition, no *ex vivo* animal model where the effects of such introgression could be meaningfully tested in preclinical models – today or ever (unless one is prepared for Jiankui He-level criminal activity on actual human embryos).

Further, and even if one assumes a miracle of a future where there is a brain organoid system where medium spiny neurons can be assessed for the “IQ-enhancing effects” of this introgression – there is highly unlikely to ever be a delivery modality to the forebrain that would allow comprehensive editing across the entire organ. This means that only embryo editing could ensure that a person with a fully edited brain is born. This reviewer is hopeful the authors agree that attempting such experimentation on embryos for “IQ increase” should forever remain beyond the pale.

Next, there is Mendel's second law to consider. Imagine a future where – via a set of technical miracles – it becomes feasible to make a human bearing IQ-enhancing variants across 20 loci in homozygous form. Given Mendel 2, after this high-IQ individual has a biological child with an unedited partner, the 20-locus-heterozygous individual resulting from that union will produce gametes with a 20-dimensional Punnett square (less so if some of the loci are tightly linked). How are we proposing to assess the safety of this manipulation (given that it will take 18 years for that F1 to have a child of their own – and we presume at least a decade for that F2 to reach an age where their “IQ” can be assessed)?

Next – what is the plan with respect to validating, specifically for “IQ,” the authors statement (line 234) that human genetic variation is additive and genetic-background-independent? Development of “cognitive ability” is not serum lipid chemistry (where a single-gene KO will be highly penetrant across all genetic backgrounds).

Finally – the paper the authors cite has the following extraordinary statement buried in Materials and Methods: “Different measures of intelligence were assessed in each cohort but were all operationalized to index a common latent g factor underlying multiple dimensions of cognitive functioning.” All of this polysyllabic verbiage can be distilled to one thing: that intelligence, like height, is a single thing that can be measured with one number. Like the authors, this reviewer is not a cognitive scientist. Having said that, debates about reification of “intelligence” to a height- or weight-like quantifiable property have been raging for a century. What is the current consensus on this “common latent factor” as being a meaningful metric of cognitive function? Is it legitimate to assess “different measures of intelligence ... in each cohort” and then act as if they are measuring the same thing?

In sum, the part of Fig 2 (and cognate narrative in the main text) is, in the eye of this reviewer, deserving of significant revision. The popular press will promptly convert Fig 2 in its current form to “Can Editing Genes Make Your Kids Smarter? Yes It Can, and This Will Happen Soon! (say leading scientists writing in the leading journal “Nature”).” This will happen if this portion of the manuscript remains as-is – surely this is not the authors' intent.

//Fyodor Urnov, Innovative Genomics Institute, UC Berkeley//

Referee #2

(Remarks to the Author)

In this manuscript, the authors make a set of calculations that indicate theoretical changes in human trait distributions and the prevalence of several major common diseases if multi-variant gene editing of gametes were undertaken at a population scale. These calculations are then used as a grounding for an ethical discussion about the pros and cons of such editing. Considering the importance and sensitivity of the topic, given past (& potentially future) atrocities motivated by eugenics, these calculations and their associated results could hardly bear greater weight. Despite this, the calculations performed here make a range of assumptions known to be incorrect, rendering the results - at best - highly unrealistic, and - at worst - exaggerated and dangerous (given the context). Below I focus on the problems that I have with the calculations rather than the ethical discussion, given my expertise, but reiterate that the ethical discussion stems from, and is thus potentially compromised by, those calculations/results (abstract: “...we use these calculations to ground a discussion..”).

The main problem that I have with the calculations/results is that they rely on a set of highly unrealistic (almost outlandish) assumptions, and while the authors do acknowledge some of these, I think worth stating that the calculations assume: (i) gene editing can be performed at 100 sites without error or off-target effects, (ii) every individual in the population undergoes the same ‘polygenic’ editing (only possible in a tyrannical dictatorship), (iii) effect sizes derived from genetic association studies are estimated accurately and are causal and are not subject to winner's curse (none of which is true, and unaccounted for winner's curse ensures that all estimates provided are inflated), (iv) there is no pleiotropy (yet we know it is the norm), (v) there are no G*G interactions (i.e. effect sizes are additive across multiple variants even when additivity implies increases in height of 8 SD units and of IQ of 5 SD units), (vi) no G*E interactions (e.g. no ‘feedback loop’ effects after dramatic changes in depression prevalence or population IQ), (vii) there is no action of natural selection (e.g. limiting such dramatic phenotypic effects; yet recent work, including by the 1st author, has shown that past/ongoing selection on

complex traits has been pervasive and critical to observed polygenic architecture).

It is common in statistical genetics studies to simulate a range of scenarios that are relatively realistic given what we know, and I see no reason not to have done that here. Results based on a range of more realistic scenarios (e.g. 1% population uptake of editing, some antagonistic pleiotropy, stabilising selection reflecting that observed in real data) could have grounded a more nuanced, relevant and urgent ethical discussion, involving issues such as: the impact on population differentiation (& society) if only wealthy individuals and countries have high uptake of editing (as is *extremely likely*, rather than merely one of two possibilities as described by the authors), how to decide which diseases should take precedence when there is antagonistic pleiotropy between diseases, is a (realistic) predicted reduction in disease prevalence worth the potential reduction in flexibility to adapt to future environmental changes (e.g. infections), and are the potential gains of polygenic editing over single/oligo editing worth the potential risks. It is a pity that the results presented are too unrealistic and cursory to inform such (needed) discussions and instead provide the opportunity for broad, and potentially dangerous*, misinterpretation. I would have expected the presented results to only be shown (e.g. in Supplementary Material) as theoretical (but implausible) upper bound results to show predictions in the absence of real processes (pleiotropy, selection etc).

(*why wouldn't one of the world's numerous dictators see these results and associated ethical defences as an instruction manual for making their population disease-free, 2-feet taller and 75 IQ points smarter than all other populations, within one generation? In practice, attempted implementation of such widespread gene editing could lead to population wipeout, but the reasons for that are not highlighted in this article, while the (likely highly inflated) 'motivation' is)

Given potential public reaction, I would personally recommend only seeking to publish this manuscript (in any journal) once the authors have performed comprehensive, realistic calculations that explore plausible future scenarios (10-15yrs from now.. which gives enough time to have ethical discussions grounded in relation to those predictions and time to update those calculations in ~5yrs time based on newer information, such as editing quality and uptake rates), alongside corresponding ethical discussion. However, in case useful, below are more specific issues that I have with the calculations in particular:

Almost no information is provided about the genetic variants theoretically edited in relation to the figures (apart from e.g. two variants described in Supplementary Box 1). It would be useful to at least provide details of the first 5 or 10 genetic variants prioritised for editing for each of the traits/diseases, giving details of their allele frequencies, effect sizes and the sources of those estimates (e.g. which GWAS/paper).

I found Supplementary Box 1 really useful for quickly checking how the authors worked out the impact of editing one variant on disease prevalence, but I think it would be useful for readers to run through the calculation relating to the first two variants edited for several of the traits/diseases (which they can then match to the figures) because once the reader knows how that is done for two variants then they can extend for any number of variants.

From reading the article one could believe that it is the 'polygenic' editing of many variants that has caused such dramatic changes in traits/diseases presented here - but really much of those changes are the result of editing one or just a few variants for which there is a rare protective allele. If this were clear, then it would also be clear that the ethical discussion here is essentially grounded on terms similar to that of the many ethical discussions published since the editing performed by He Jiankui of a HIV-resistance variant (i.e. that editing of individual rare protective alleles can dramatically modify individual/population risk).

Interpretation would be improved if the X-axis included 0 edited alleles (trivial for traits, but especially for disease prevalence this would make clear the change that just one allele is edited Vs subsequent edits). I don't think it is appropriate to display prevalence (Y-axis) changes on a log-scale, which exaggerates the impact of editing many, compared to a few, alleles. Making these proposed changes to the axes would help to address the point made in the previous paragraph.

The results presented are population-level (changes in trait distribution / in disease prevalence) and yet at various points throughout the ms the authors talk as though they have made individual-level inferences, e.g. in the abstract "Editing a relatively small number of variants can make a substantial difference to an individual's risk of developing coronary artery disease, Alzheimer's disease..". Clearly if an individual is homozygous for the APOE-e2 allele then the proposed edits will not lead to a reduction in disease risk corresponding to the reported reduction in disease prevalence. The variance in individual changes in traits and disease risks has not been considered here whatsoever, but would be interesting to estimate and present and would allow comments in relation to individual-risk.

The authors state in the abstract "Our modelling shows quantitatively how putatively positive consequences of gene editing at an individual level may deepen existing social inequalities". This seems an unusual statement given that no part of the modelling performed considers potential impacts on social inequalities. I think it would have been really interesting to investigate this, but given that no such modelling was performed, the article effectively leaves the impact on social inequalities open to widespread/public interpretation, which I think is risky given the potential stakes here.

Stabilising selection on human traits has helped to shape observed genetic variation and the genetic architecture of traits, presumably including selection against certain high-order combinations of alleles present in the population. In fact, this is nicely demonstrated by the 1st author of the present manuscript in Sanjak et al 2017 (PNAS, doi: 10.1073/pnas.1707227114), in which it is stated "for several traits, we demonstrate that individuals at either extreme of the phenotypic range have reduced fitness", one of which is height (see Fig.2). Thus, I find it surprising that the 1st author

considers it reasonable to present results that ignores the likely impact of such selection, inferring that genetic editing of multiple alleles could result in a 8 SD increase in height in the population (even without considering the implicit physiological birth-related issues). The results from Sanjak et al 2017 (see Fig.2) could in fact be used to make the case that even a small number of height allele edits could reduce fitness/fecundity and thus the population size due to selection (or at least in those undergoing editing).

The fact that the calculations predict almost complete eradication of depression in the population on editing 50-100 genetic variants seems to highlight the disconnect between the calculations/results and the likely interplay between genetics and the environment (as well as other limitations of the calculations discussed above). Even if baseline genetic risk for depression is modified dramatically in all individuals, factors such as societal position, interactions among people, competition etc, would surely result in depression at substantially higher rates than predicted by assuming that modification of genetic variants can impact risk in isolation from the environment (/resulting environment).

It would have been nice to see some investigation of real data, to test for enrichment or deficits of individuals with combinations of the alleles proposed for editing here (indicating viability / pleiotropic effects / selective pressures).

“Human enhancement” is rather a loaded phrase and is subjective - an explicit assumption is made here that increases in height/IQ corresponds to human enhancement, but I can think of a few arguments against this claim. Also, the controversy around the meaning of IQ and the genetics of IQ are ignored here - I cannot speak to the former, but in terms of the latter it has been clearly demonstrated that much of the previously estimated heritability of educational attainment is due to the ‘household effect’ (i.e. is due to the environment created by parents), and I suspect similarly for IQ, while some of the remaining heritability will be explained by G*E correlation, and so the identification of G-only causal effects on IQ may be particularly challenging.

While the authors claim that PGE could have a far larger impact than ESPS is true in theory, I think the opposite will likely be true in practice. ESPS is presently legal in many jurisdictions and can become substantially more efficacious with relatively minimal technological advance (generation of many more embryos).

Referee #3

(Remarks to the Author)

Visscher, et al. have written a prospective piece on human germline genome editing that contains a good amount of nuance but ultimately comes out in forceful support for the idea that we should eventually consider editing embryos with the aim of altering complex traits. If published, the piece is sure to be extremely controversial—I’d guess it will dwarf the controversy that surrounded the publication of Harden’s Genetic Lottery (a text that should probably be cited when the authors reference a “genetic lottery” in line 355).

The first section of the perspective invites us to consider a world in which we are capable of precisely editing a human embryo at tens of loci, and in which we have fine-mapped human complex traits successfully enough to know where to make edits to most effectively perturb traits like diabetes risk, height, and IQ. Within the context of this model, the authors calculate how much we might perturb various human phenotype by means of editing a particular number of loci. The second section of the paper then explores ethical pros and cons that should be considered before deciding whether to apply such a hypothetical gene editing technology at scale.

The concrete predictions that are made in the first section of the paper are bolstered with some technical supplementary information, but I did not find this technical supplement to be detailed enough to judge the correctness of the numerical claims made in the text and in Figure 1. The supplementary note is clearly written, but it is extremely succinct, and it omits definitions of many necessary technical terms (including f_0 , λ , K/K_g , ϕ , and others). These omissions aside, I think that this mathematical framework seems basically sound in that if you input variant frequencies and effect sizes, you can accurately calculate the effects on the population trait distribution that will result from editing these loci. However, the paper includes no details about what concrete frequencies and effect sizes they plugged into this model to obtain the predictions in Figure 1, which claims that editing tens of loci should be enough to change various trait means by multiple standard deviations. These effect sizes seem extremely large in light of the modest amount of trait heritability that is explained by published GWAS, and should not be published without a more detailed accounting of the parameters that were used to derive these results.

The authors do cite particular genome-wide association studies that they presumably used to obtain the allele frequencies and effect sizes that yielded their predictions, but several of the studies they cite are affected by problems that the authors do not address in the paper at all. For example, they cite GIANT consortium association studies for height and BMI, but do not mention a pair of papers (Berg, et al. eLife 2019 and Sohail, et al. eLife 2019) that showed that the GIANT consortium likely severely overestimated these trait associations due to population stratification. This stratification problem also affects the conclusions of Field, et al. 2016, which is cited in support of the idea that these traits are rapidly evolving. If the authors used beta values from the GIANT papers in their model, they are likely overestimating the impact that editing a few loci would have on height and BMI. It is also likely that GWAS of traits other than height and BMI are subject to as much or more confounding by population stratification, meaning that all of the effect sizes output by the authors’ model are likely to be overestimates. I don’t expect the authors to correct for this systemic effect, but it should be mentioned so that readers are aware that these variant effects may be overestimates. Even if we take the results of the GIANT consortium at face value, I am still puzzled by the paper’s assertion that editing ~20 BMI-associated alleles is enough to reduce BMI by a full standard deviation, given that the GIANT consortium found that 941 genome-wide significant SNPs only explained 6% of the total population BMI variance. The authors make no mention of releasing code or the table of data that went into Figure 1—without such data, it’s not possible to judge whether these results are reasonable, especially when the magnitude of the

results seems so high compared to estimates of trait variance explained in the cited studies.

One stylistic choice that I found extremely problematic was the discussion of editing height-associated and BMI-associated loci to achieve unidirectional “desirable” changes in these traits. Although the authors acknowledge that extending trait values beyond their natural ranges could cause problems, they still seem to take for granted that everyone would prefer their children to be as tall and thin as possible, which is frankly bizarre and stigmatizing to individuals and populations that are shorter and heavier than others. The discussion of these traits does not even square with the paper’s thesis that gene editing should be used to reduce morbidity and mortality, given that height has no effect on mortality and that “overweight” individuals actually have lower all cause mortality than “normal weight” individuals (Flegal, et al. 2005). In line 203, the authors refer to high BMI as a causal risk factor for various diseases, but do not note that this causality is debated—for example, much of the decreased disease risk that is sometimes attributed to weight loss better attributed to diet and exercise changes that sometimes but not always cause weight loss (e.g. Ross and Janiszewski 2008). This confounding takes away credence from the authors’ assertion that genetic modifications to BMI should cause the same disease risk reductions that environmental weight loss interventions produce.

The assumption that higher intelligence is always more desirable is perhaps less eccentric than the papers’s assumptions about the desirability of height and thinness, but it is still stigmatizing to individuals with lower IQs. In addition the statement about selection on loci associated with educational attainment (line 646) is a bit misleading, since most of this paper takes for granted that more intelligence/educational attainment is better while the cited sources claim that there has been selection *against* variants associated with increased educational attainment.

The paper could make all the points it is trying to make without perpetuating as much stigma if it restricted its scope to the idea of reducing disease incidence without taking for granted Western culture’s assumptions about which anthropometric traits are “desirable” and “undesirable.” It would not be a bad idea to expunge “desirable” and “undesirable” from the paper’s lexicon.

The ethics-focused portion of the article does a reasonable job of articulating the main debates around the advisability of germline genome editing, but I thought the ethical implications of off-target editing errors deserved a bit more discussion. I realize that the authors are talking about a hypothetical future in which editing error rates are extremely low, but I don’t think it’s reasonable to assume that editing will ever be error-free. This means that for editing to be acceptable, we need to decide that the benefits of editing outweigh the risks of harming the embryo. Would it every be acceptable to change a cosmetic characteristic like height at risk of introducing mutations that are harmful to health? A similar calculus needs to be applied to the discussion of pleiotropy and the uncertainty of genotype-phenotype associations—one could argue that editing a genome to introduce a rare variant is never ethical because the rare variant has not been tested on enough genetic backgrounds and might introduce a deleterious effect that is not known from genome-wide association studies. I don’t agree with the assertion in line 235 that experiments have shown most genetic effects to be additive—I think a more correct assertion would be that when alleles are rare or effect sizes are small, an additive model works to a first approximation, but this would not apply if we used editing to suddenly make a rare allele of large effect into a common variant (as in the discussion of using rare “protective” variants to lower cholesterol). In addition, although the table of arguments for an against PGE is reasonably useful and comprehensive, it contains some sound bites that are pretty unsavory, e.g. “not all forms of diversity are valuable” and “human bodies have clear design flaws.” Who gets to decide what kinds of diversity have enough value to be allowed to exist, and classify a feature of someone’s body as a “flaw”?

I found the discussion of eugenics reasonably complete and balanced, but it also comes rather late in the article given that eugenics will be many readers’ first mental association with germline genome editing. It could be a strategic choice to move much of this material into the introduction of the paper. On a related note, in line 431 the authors cite the idea that the human gene pool is accumulating increased genetic load due to modern medical care, and if they cite this idea, they should acknowledge that it is highly controversial (e.g. see Roth and Wakeley 2016). I would also take issue with the statement in line 455 that we have an “obligation to to manage genetic risks [sic] factors”—the idea of “managing” the human gene pool in any kind of top-down way is a eugenic idea that compromises the ultimate freedom of individuals to decide whether and how to reproduce. I would urge the authors to at most recommend that some forms of PGE be made available for individuals who wish to use it and refrain from the implication that anyone should ever be “obligated” to make use of such technology.

Referee #4

(Remarks to the Author)

I think this is a useful piece. We’ve been talking and writing about these issues for over 50 years but looming reality, like the prospect of hanging, has a wonderful way of concentrating the mind. (Though you might want to note a bit more the age of the debate on genetics and throw in a few more cites to the pioneers of the discussion.) Your piece should force people to take these issues more seriously and to focus their attention on important “details,” like possible regulatory schemes.

I do have one major reservation about it as well as a number of smaller comments.

My major reservation is that your conclusions about the size of the effects are so startling as to border on unbelievable. Increases of 75 IQ points or 50 centimeters in height (sports would certainly be different if the tallest people were 9 feet tall with adjustments needed to basketball nets, soccer goals, and volleyball nets among other things!) seem fantastic.

I do not have any expertise that allows me to understand, let alone criticize your model. I do hope the editors have sent this paper to some experts who can. If such reviews lead the editors to conclude that the projected magnitudes of the changes you foresee are outlandish—not just in terms of issues with your particular model but from limitations in the power of PRS—then I think this should not be published. If, as I rather suspect, such reviewers would say something like “these estimates seem very much on the high side but it is plausible that some very large effects might (or might not) follow.” In that case, I would recommend publication.

That's out of my hands (and expertise). But I do think that if that verdict allows the paper to go forward, it still should have even more caveats about the size of the impacts. You have some, but should have more, perhaps along the lines of pointing out although your estimates might be at the high end of plausible projections, that even if the increases are "only" half as large, they are very significant. Your paper's impact comes from the idea that this technique could make big changes but even if someone came away from the paper thinking there is only a 20% chance you are right, or even close to right, about the scope, that should still motivate increased practical attention to these issues.

Now for some other comments, ranging randomly from small to large.

You cite the George Daley et al NEJM paper on paths forward. I think you should, in the same vein, cite the reports from the WHO and NASEM/Royal Society working group/commission.

Make it clearer that Aurea was selected, not edited. You say that initially but by the end of the paragraph, I worry that that nuance will be forgotten. In general, I think the piece should emphasize even more how PGE is different from PRS PGD.

I do think you are very much on the bullish side on the power of PRS. And the fact that you are citing a 2018 paper for its big claims that PRS can (often be?) as powerful in its predictions as those for monogenic conditions actually makes me nervous. That's five years old but the controversy in the field is (as far as I can tell) still raging. (I do like the point about rare protective alleles, where I can see a strong effect as quite plausible.)

When you get to the limitations in the model (which, to be fair, seem to me to be major limitations in all the models showing strong effects for PRS, not just yours, I think you should change their order. The two most important limitations are the ones you list fourth and fifth: epistasis and environment, followed closely by pleiotropy. The first two are largely technical (although I do suspect that more samples leading to more "causal" variations will also increase the significance of the problems of epistasis and environment. I'd take them in order of hardest to easiest, in the spirit of putting the biggest risks first in the consent form. (By the way, I'm not sure epistasis exactly is behind my own suspicion that models make too many assumptions about how the various polymorphisms interact, often by assuming they are additive when there is no particularly reason for them to be. But I can't even tell if your model is an additive one.)

I didn't like "Other countries, notably the US, adopted "negative eugenics". Lots of countries did so, including Canada and much of Northern Europe, especially, of course, Nazi Germany. Later you include some more countries, but it seems to me unfair to single out the US in your first mention on negative eugenics.

I don't disagree with your statement that "previous eugenics practices...were clearly unethical", but you might want to give a reason that at least forced sterilization was clearly unethical, presumably the coercion involved.

I'd add to the response argument for enhancement that we already are very different from past generations, obviously in cultural ways but also in biological ones.

The "non-identity argument" in box will puzzle non-specialists. I think you need to explain it more clearly (and good luck doing so in only a few words).

You talk about the expressivist argument and its effect on those with disabilities and both the argument and the response are fair, but I would also add that the elimination or great reduction in disease incidence will have more concrete effects on people with disabilities—less research, fewer specialists, less social support. (Personally, I think the disability issues are the hardest, for me, in genetics.)

You talk about surveys of "people in most regions" but how many regions? How many of them were sampled? And how fully? I suspect most of the world's cultures and peoples were greatly underrepresented, but maybe not. Some specifics would help.

The argument on the difficulty of drawing the line between enhancement and treatment has never seemed strong to me. Yes, there will be fuzzy zones. No, the vast majority of applications will not fall into those zones. And we have ways of drawing, imperfectly but usefully, lines in fuzzy zones.
argument isn't very strong

I don't like your AD example. I suspect simple multiplicative combinations won't work for it because the odds of having each of those ten won't be independent of each other (even if such alleles are real and work in something like an additive manner, as to both which I would need to be convinced).

I'm also troubled by your "populations may benefit sometimes from some prevalence of a harmful allele" argument, not because I don't believe it but because if we learn THAT much about the genetics of common disease, we'll be able to treat it even (especially?) in people at high genetic risk. Medicine is part of the relevant environment and it should advance at least as much as our understanding of the meaning of GWAS and PRS.

By the way, is there any reason to think that there are any alleles with solely positive effects? And if there were, how in the world would we know that—proving a negative is going to be hard, especially if that negative means showing it might not be harmful in the face of a new pathogen or new environmental factor.

I really hate the “genome degrading” argument, and not just because it is so reminiscent of the early 20th century eugenicists (“The Death of the Great Race, etc.). As you know, and note, the significance of genetic variations (and phenotypic ones for that matter) depend on the environment. Insulin undoubtedly led to a higher percentage of people having risk alleles for type 1 diabetes because now they could survive to reproduce. So what? If we can treat type 1 diabetes (or, to be fair, if we could treat it perfectly, which we can’t now but should be able to do before you can do PRS editing), why does that make the genome degraded? We have undoubtedly lost (or had lowered incidence of) some alleles good for being a hunter/gatherer. So what? And the word “degeneration” is just creepy. You should eliminate it.

Having said that, I do like (mainly) your argument for a more collectivist approach, although spelling that out is probably more the work of more than one book.

I wish you would spare a thought, and a paragraph, for the practical ethics of starting down this path. Who is the first PGE’d baby? How safe and effective does it have to be before we are willing to risk a baby’s life and health on it? And how would we know how safe and effective it was? One good reason for dealing with awful diseases is that the risks of non-intervention are often great enough to justify some unknown risks of intervention. PRS will normally involve smaller benefits, especially if medicine continues to progress.

One last idea – when discussing massive editing, you might throw in a sentence about genome or chromosome synthesis. In 20 or 30 years, it may be cheaper and easier than making 2000 edits.

So, overall I’ve got a lot of comments that I hope you will consider, but few that I think really must be incorporated in a revision. I enjoyed reading the piece (except for the model, which I didn’t even attempt to read.)

Version 2:

Reviewer comments:

Referee #2

(Remarks to the Author)

The efforts that the authors have made to incorporate some more realistic modelling scenarios have helped to ground a more informed ethical discussion. However, I think the manuscript still has a number of issues that should be addressed, especially given the likely attention and possible consequences of the paper.

Evaluating the paper as it is now, I think the statistical genetic modelling demonstrates two key points that - despite being simple and intuitively clear even with little/no modelling - most people in the field and beyond may not have considered:

(1) the editing of a small number of alleles (< 10) could have a dramatic impact on individual risk of major complex, polygenic diseases and the values of complex traits.

(2) as more alleles are edited for each disease there are diminishing ‘gains’ and increasing risks (off-target effects, antagonistic pleiotropy, severe effects of stabilising selection, reduced diversity), which at scale in a population could lead to catastrophic consequences (large-scale death, prevalent unviable foetus, major population reduction or wipeout).

I think these points could have dramatic consequences for the future of medicine and society and do represent the grounding for an important ethical discussion on the topic that could be valuable to initiate now. However, I think the ethical discussion should be framed more in relation to the spectrum that these two points highlight - i.e. that editing a small number of alleles has the potential to reduce risk of major disease(s) in edited individuals with incurred risk that may be outweighed by the benefits (analogous to medication and its side effects), but that editing a large number of alleles is likely to produce risks to the individual and population that outweigh the benefits [Note: my comment here relates only to statistical and population genetics inference, not to any ethical issues]. Or to put it another way, “oligogenic editing” of genomes (< ~10 edits) may have clear health benefits to individuals that society should at least consider seriously (inc. the ethical implications) given the goal of healthier populations, while “polygenic editing” of genomes presents likely catastrophic consequences for both individuals and populations (of the scale of the potential consequences of climate change, nuclear power and unregulated AI). While there’s no obvious line between oligogenic and polygenic, I think it’s clear that any predicted benefits from editing become serious dangers somewhere along that continuum.

The authors do highlight that “editing a relatively small number of genomic variants” (abstract) can reduce disease risk substantially, and all their main figures now relate to 10 or fewer allele edits, but throughout the ms the authors talk about the effects and ethical implications of “polygenic editing” (including in the title). This conflation causes unnecessary problems, because polygenic editing should be able to be ruled out as a clear danger by the authors (barring unforeseeable changes) on the basis of present knowledge and their own modelling, whereas the real modelling and ethical uncertainty is whether oligogenic editing could be more of a benefit or harm. Moreover, oligogenic editing is far more likely to be socially acceptable than polygenic editing (which I think is similarly unlikely to be socially accepted as ESPS of 10,000s of embryos).

Some other issues relating to the statistical genetic modelling and results:

- For the main results presented, the authors chose not to model the more realistic scenario that only a very small fraction of

the population would perform PGE. If there was an e.g. 1% uptake (perhaps on this high side), then there would be almost no change in the population prevalence of these diseases, and although results/text refers to 'edited genomes', I think this point should be made crystal clear to the readers so that they understand the likely (lack of) impact of future genome editing on common diseases in populations (unless rolled out by national healthcare systems, which seems unlikely).

- For the disease plot of Fig.1, the Y-axis should show fold reductions in increments of 1 between 1x and 10x (2x, 3x, 4x, etc) since these are where most of the 'action is' here but the reader will be unable to easily estimate what fold changes these correspond to.

- The new section at the top of "Model limitations.." about potential advances in somatic editing obviating the need for heritable editing, seems out of place here and would surely be better suited to the introduction or at the end of the paper. The link here is with 'future environments' as though addressing G*E effects, but this link seems highly tenuous.

- The modelling of G*E effects, as presented in Fig.2, is rather simplistic in relating to assuming that future environments will simply change traits by some degree. More pertinent G*E effects are those that could occur after dramatic changes to traits, given the balance created between observed genetic variation and the environment caused by selection (i.e. G*E effects with highly deleterious results due to genetic profiles poorly matched to the present environment), or those that mean that diseases such as MDD may operate in the population in a relative rather than absolute sense (i.e. prevalence remains stable, despite the inferences of Fig.1, due to prevalence being more due to relative risk than absolute risk). I think the possibility of such G*E effects should be at least mentioned here.

- L258 "the effects inducing large changes associated with most polygenic traits are largely unknown" - there is now a sizeable literature suggesting that most polygenic traits have been under stabilising selection, which would indicate that large changes in their values would typically result in deleterious effects. The authors should make the readers aware of this. While the authors model stabilising selection as presented in Fig.3, they do not explain that evidence suggests that most polygenic traits are under stabilising selection and that this likely restricts the extent of editing that is viable and consistent with health in the present environment ('present' on the scale of centuries).

- When the authors refer to 'reduced fitness' I think best to emphasise (for more lay readers) that they're referring to large-scale suffering and death, since this is how the consequences of present complex diseases are described (justifiably) when highlighting the potential benefits of editing.

- Enhancement section: as raised previously, given scientific debate about the meaning of 'IQ' and about indirect (/household) and non-cognitive effects in relation to Education GWAS, and the links between 'IQ' and past eugenics, I'd highly recommend dropping or at least limiting references to these in the paper. References to human 'enhancements' (also in the abstract), in relation to IQ and height (despite being dropped from the modelling), seem ill-judged - how can 'enhancements' of such traits be defined in the context of evident stabilising selection and pleiotropic effects (and given personal subjectivity in terms of ethics)? (unless by enhancements the authors mean edits that make individuals closer to the average).

- While Fig.3 indicates how inequality due to editing changes with uptake, it shows this in relative terms, which obscures the degree of inequality itself. Also, the prevalences (K) and impacts of editing (K') are for hypothesised diseases. I think it could be even more informative to readers (at least as a replacement of one of the 4 plots shown) to show this relationship in terms of absolute inequality of disease risk for the real diseases of Fig.1 based on the effects of editing 10 variants for each disease - this would show the degree of inequality (e.g. as a variance of risks) across uptake and would also show the relative inequality across the different diseases (not observable when standardised as now).

Referee #3

(Remarks to the Author)

This perspective has been much improved by the omission of height and BMI as example traits and the increased focus on possible deleterious consequences of embryonic gene editing. The transparency of the model has also been improved by addition of source code and additional methodological detail. Overall, the work still seems pretty likely to spur controversy, but perhaps a more productive controversy than would have been the case in its original form.

There are still some places where the article expresses a viewpoint that feels dangerously close to eugenics: the view that society should collectively decide to eradicate certain undesirable genes and encourage treatments that will nudge society toward a genetic profile that is optimized for producing some notion of a good life. I think this problematic stance could be tempered by adding additional emphasis on the role of individual choice: in my view, scientists and politicians have a responsibility to decide that certain gene editing treatments should never be developed because they would have a harmful effect on society, but the converse is not true: they do not have the right to recommend that certain gene edits should be broadly adopted. We see this distinction with the way that prenatal genetic testing is offered to patients today: scientists and doctors have decided that this testing is enough of a social good that it should be routinely offered to patients, but it would not be ethical for anyone's doctor to pressure them to consent to such testing.

I think that the three paragraphs starting on line 473 get a bit too close to implying that people should one day be pressured to consent to gene editing in the way that doctors currently pressure people to quit smoking and lose weight. It's not clear how far the authors are thinking we should go in having "substantial ethical reasons to use the technology to prevent the transmission of these genes," so it would be best to clarify that we should only go as far as offering such editing to

enthusiastic customers.

Another critique of this section is that it could be problematic to use “overall risk of death” as a criterion for offering gene editing. After all, a gene associated with darker skin or non-conforming gender identity probably increases the risk of death in a society that is biased against these traits, but that does not seem like a good reason to encourage editing of such loci.

The role of individual choice and agency should also be added to the section “Moving forward—the need for collectivist ethical approach to human genome editing” to emphasize where collectivist imperatives end and become trumped by individual choice. Specifically, individuals have the ultimate deciding power about what they think constitutes a good life and whether that makes them want to avail themselves of currently available gene editing technologies.

I take the authors’ point that the term “human enhancement” has precedent, but it still seems like a more neutral term like “attempts at human enhancement” or “cosmetic alterations” might be preferable to emphasize the subjective nature of such efforts.

The discussion of SES is overall pretty thought provoking and nuanced. One additional point that occurred to me is that low-SES individuals might feel more pressure than high-SES individuals to avail themselves of gene editing technologies in the same way that they can feel extra pressure to strong-arm their kids into studying hard and pursuing high-earning careers.

I found the discussion of pleiotropy to be much improved, but one important thing that is not mentioned is that rare protective alleles are especially likely to have epistatic consequences we are not aware of since we have not had a chance to observe them in combination with very many other rare alleles, and natural selection has also not had the chance to select against any deleterious effects that these alleles might have in combination with various other rare variants. It seems likely that any attempt to make rare alleles more common should be preceded by some kind of variant effect scan that attempts to test how these alleles behave on a variety of backgrounds and anticipate deleterious epistatic effects.

Referee #4

(Remarks to the Author)

As a result of your responses to the reviewers, this paper has become both stronger (more publishable) and weaker (in its implications). And that does seem to me the right result.

Assuming the scientific reviewers accept your changes (the validity of which I am in no position to judge), I think it would be a useful contribution to the literature. I do, of course, have some comments. Three of them are general but most range from small to trivial (typos). I’ll start with the big ones.

General Comments

First, I think your responses to the reviewers have introduced substantial uncertainty into the question of the power of PGS and to its risks. This seems to me necessary, but I do think entails some more changes in the article. The current version is not quite as strongly “this will happen” as the earlier version, but I think its overall tone does not yet reflect the increased uncertainties of its text.

For example, in your concluding comments you say

“over the next decades, it will become possible to” That will, at least to the extent it implies “safely and effectively”, needs to be weakened—to “it may” or “it is plausible that” or even (though I personally wouldn’t say this) “it is likely that.” But not “it will.”

I think you should explicitly, and up front, state that whether PGS will make sense in light of the uncertainties about its advantages and risks remains unclear. But because it is at least plausible (possible?) that it will work and its effects may be very large, it requires attention well in advance of efforts to use it.

Second, it occurred to me this time that your foreseen world, with widespread adoption of PGE, will require the almost complete abandonment of natural conception. I have myself previously argued that in vitro gametogenesis plus whole genome sequencing will lead a large percentage of the population to avoid monogenic conditions by using PGT (preimplantation genetic testing, in this context – your use of PG for polygenic could lead to some reader confusion). But that prediction encountered a lot of skepticism. I think the needed very wide transition to clinic conception should be noted somewhere in this.

Third, you end by saying “we” must use the time to figure things out. But who is the “we”? I’m not asking for a roadmap but something along the lines of discussions leading to public debates and ultimately to governmental (?) decisions, at the national and (or?) international level. (A line noting the difficulties in that, especially internationally, would not be out of place.)

Smaller Points

45-50 The discussion of Aurea seems awkward. You may need a half sentence to say why she is relevant to your

discussion.

70 You are using PGS again (which I don't like because of the possible confusion with PGT, PGD, etc.), but what happened to ESPS? Wasn't that the new acronym/term you made up for your suggested process?

82 It IS possible to do functional test for phenotypes in vitro or it may, in some circumstances, be? Doesn't that depend on the phenotype? If it is "protein X is being produced at level Y" it seems possible (though it may not occur in a full organism because of effects from other cells) but if it is, say, AD, you can't. I'd weaken the statement.

120 This line is ambiguous. I'm not sure what you mean by "reducing the lifetime prevalence by 5 to 2.9%." You seem to be saying total prevalence, but I wonder about that, as lifetime prevalence of 5% AD seems awfully low. I suppose you could be saying "reducing the prevalence by somewhere between 5 and 2.9 percentage points" (from, say 15%) or even reducing the prevalence by 5 to 2.9 percent, which, with a 15% lifetime prevalence, would only take it down to 14.25 to 14.5%. I doubt you mean the latter two, but the sentence isn't clear.

Also, does this involve editing out APOE4 alleles (possible certainly with heterozygotes) or replacing them with APOE2 alleles? And maybe replacing APOE3 alleles – because if you want to lower AD risk, you'd get rid of those, too. More clarity may be in the supplemental materials but a statement in text would help.

129 Of course, the long term effects of switching from common alleles to rare protective ones isn't well understood because we rarely have much data on the lifetime effects of the rare alleles on people. Even the CCR5 knock out effects are based on awfully limited data.

136 The "average genome" of "the population" – which population? This average genome will be different in different ethnicities, which should lead (I think) to different degrees of reduction in different populations, no? You do come back and make a point about different population at 279 but I think you should note it earlier.

175 It's not just gene therapies. More traditional treatments may also be effective, like the drugs that treatment 90% of cystic fibrosis patients now. It's not "one and done" like the gene editing (somatic or germline) would be, but whether one treatment or the other is "better" depends on lots of things including risks, cost, and convenience.

177 Since you wrote this, the first CRISPR therapy approval HAS occurred in 2023, in the UK.

181 Typo – change "if" to "it"

232 Is it "reasonable to assume"? Personally, I'd say "not unreasonable to assume"...a touch weaker. We know so little.

246 Define or describe winner's curse – you use it only once, without explanation.

264 It would be good to highlight the huge uncertainty, and difficulties of understanding possible prenatal effects on gene changes.

289 typo – edited for editing

293 typo -smaller for small. Also, I don't really understand your fetal hemoglobin example. I don't understand why the treatment has such a big effect and but the allele has a small effect. Is the changed allele extremely rare (or even non-existent) in human populations? Whatever the truth, the example doesn't make sense to me (and so might not make sense to other readers).

306 Well, some of Europe did (though not the southern and predominately catholic parts), some Canadian provinces did, and there were efforts (though I think no government action) in Australia.

313 Well, it's still to control what kind of people exist, but with the decision at the family level and not the societal one. (Although there may be some tension in you putting emphasis on this in a paper where you end by calling for more collective thinking, which some might conclude should include encouraging or requiring this kind of editing for the collective good.)

Ethics Box

Why not question whether new phenotypes would be bad – are all differences are bad? If you are listing counterarguments, I'd make that one.

Typo in "means matter" – overlook for overlooked

335-6 Hmm, height and IQ show up. I thought you took them out. Seems like you are trying to sneak them back in but through a side door, without actually "showing your work." I'd either leave them out entirely or mention them only a weak way: "It is even conceivable that, at some point in the future, traits like height and intelligence might be subject to large scale changes."

350 You talk about "concerns" and take them seriously, but "concerns" aren't necessarily "arguments." Taking them seriously politically may be necessary but do they deserve to be taken seriously intellectually without more?

372 This seems to me repetitive of the sentence at 369.

416 You might want to refer to differential health risks already by SES or ethnicity, such as huge black/white health disparities in the U.S.

434 Equitable access “to new technologies, such as GWAS today”? What access is currently needed to the “technology” of GWAS? I suspect you’ve got something specific in mind; if so, say it.

449 The “it depends on how it’s regulated” take seems inconsistent with the problems of international disparities, which you highlighted four lines earlier. Fixing that would take a lot more than good regulation or non-market provision. You might acknowledge the special difficulty there.

453 I wouldn’t use the heading here “Pleiotropy and Genetic Diversity.” Table 1 mentions diversity (VERY briefly) but not pleiotropy. I think this header might confuse a reader, if only momentarily.

473 This paragraph seems very different from the rest of the paper and feels out of place. Maybe it needs some more explanation to fit it into your flow. BTW, is there much work on PRS for overall risk of death? If so, it might be good to mention it.

505 typo – that do not

510 “Genetic load”. You state, without citation, that health care is causing mildly deleterious mutations [do you mean that, or alleles?] to accumulate.” I believe that must be the case but is there evidence that it has actually happened? I presume the models referred to in the next sentence are mathematical models, not based on actual data about known increases in, say, type 1 diabetes predisposing alleles.

On this whole argument, even if the genetic load argument is right, those future generations could presumably make up their own minds about ways to improve their health, through your PGE, through PDT, or otherwise. Arguably you are doing them a disservice by limiting their choices. (Assuming they will have choices, which I think you do.)

520: I know this whole new section responds to reviewer comments. You could tailgate here on the National Academies’ group’s ideas on a path toward what it calls HGGE...in fact, you are recommending that the efforts start with HGGE.

522: The sentence makes it sound like Tay-Sachs editing would be PGE, but it’s really HGGE. I’d say “We should first evaluate the use of embryo editing, sometimes called human germline genome editing [if you don’t use that or something else for it earlier] for”..Tay-Sachs. Similarly at 532, the “it” that could be extended isn’t really PGE, just HGGE. Basically, this would be a lot clearer if you just said that PGE’s pathway would necessarily follow on, and after, success with the HGGE pathway.

And, of course, first would really be preclinical work, in vitro with human embryos and in vitro and in vivo with non-human animals, especially non-human primates.

And to the extent you want to use rare protective alleles (which you clearly do), I think you should call for more research on the consequences of those rare alleles on total lifespan health of people who naturally carry them.

540 You say “children” screened at birth but I think it’s still just “at least on child.”

546 Once again, you sneak back to height and IQ. See my comment at 335.

I think that’s it – more than enough, I’m sure you feel. Obviously, some of these are more important than others, especially on issues of style. I hope this helpful. I do think —if this passes muster with the scientific reviewers —this would be a useful article.

Version 3:

Reviewer comments:

Referee #2

(Remarks to the Author)

This manuscript is now vastly improved from the original. I am satisfied with the latest round of responses and any remaining points of disagreement are essentially subjective/stylistic in nature (in particular about it being most worthwhile to focus on and discuss the editing of a small number of variants, rather than on more ‘polygenic’ editing, which seems to me worth effectively ruling out in the short/medium term).

One point though - line 20 in the abstract reads "Editing a relatively small number of genomic variants can make a substantial difference to an individual's risk.." - should change "can" to "could" since it is not known for certain (at least not

from modelling).

Referee #3

(Remarks to the Author)

The manuscript is much improved--at this point, it seems likely to spur healthy debate rather than unhealthy amounts of controversy. The authors have done a great job incorporating all of our diverse feedback while still making concrete quantitative points that will add to the discussion on this topic.

Referee #4

(Remarks to the Author)

So, authors, have you second-guessed during this process whether you should have ever started this paper? I want to compliment you for sticking with it and for responding so well to the many comments made by your reviewers. Your responses seem to me to range from convincing to adequate (meaning "ok but not the way I would have said it"). Overall, I think the comments from the reviewers (especially the scientific reviewers) AND your responses to them have improved this paper markedly. And I think this is (almost) ready to be published.

I have only one last (two pointed) comment, which I had earlier tagged to line 520, about the National Academies/Royal Society efforts. I accept your argument for why your approach is importantly different, and, with understanding but some regret, your view that explaining the differences would be too distracting.

Point one - think again about whether explaining the differences would take you too far afield. In the responses to the review, it took you only about 10 lines to explain them well. I bet you could get it down to 5 or 6 and I think it would worthwhile...as at least some, sophisticated, readers will say, "what about the NASEM/RS approach, which they don't even deign to mention. Did they not read it or are they avoiding it for other reasons?"

Point two - even if you don't talk about the NASEM/RS approach, you should at least cite it somehow, if only to make it clear that you DID notice it. (Plus, it's polite...and, note, I wasn't involved in that report at all, as far as I can recall.)

Otherwise, a very nicely done job.

Referee #1 (Fyodor Urnov, Innovative Genomics Institute, UC Berkeley). Expertise: gene editing for human disease

As someone whose daily practice for the past 2 decades has been building genome editing as a tool for clinical intervention, this reviewer was prepared to be concerned about the submitted manuscript as distracting from the immediate cause of expanding the public health impact of CRISPR on existing disease. To the contrary, the manuscript is a thoughtful and cogent extended argument in support of the authors' statement (line 80) that "it is vital to discuss the practical, ethical and social implications of PGE now." The work is timely and of broad interest, and the authors are to be commended for taking on such a thorny subject.

We thank the referee for their time spent on reviewing our paper and for their positive and constructive comments and suggestions.

From the standpoint of this reviewer's area of immediate technical expertise the major issue – that can be addressed – of the submitted manuscript is a certain level of separation from the actual clinical practice of genome editing today. Suggestions for manuscript improvement to that effect are below. Separately, a significant concern – detailed below – is the authors' discussion of PGE for IQ enhancement. In the eye of this reviewer, this passage requires significant revision.

We thank the reviewer for these comments. We agree that what we discuss in our paper is (far) removed from the clinical practice of (somatic) genome editing today. We have watched the presentations and discussions from the recent Third International Summit on Human Genome Editing online (<https://royalsociety.org/science-events-and-lectures/2023/03/2023-human-genome-editing-summit/>) and concluded that most current efforts are on single gene disorders/diseases and that off-target effects and mosaicism are (still) a concern. There is also exciting progress on new genome editing technologies. This field of research is not ours but there appears to be an incredible pace of new discoveries and new gene editing approaches, in addition to multiple clinical trials and the first CRISPR therapy for SCD and beta-thalassemia seeking FDA approval.

Following this reviewer's comment and that of others, we have now dropped the example of IQ altogether from the paper. We have also dropped BMI and height as example traits and focus solely on disease and their risk factors.

61 and elsewhere: as the authors are doubtless aware, the CRISPR-based approach that is closest to approval as a prescribable medicine is editing a GWAS hit, BCL11A (30355263). Of direct relevance to their argument is the fact that the natural allele of BCL11A is weak and changes HbF by ~0.5 g/dl (out of a total of 15 g/dl) – ref 18245381. The edited allele – as measured in gene-edited patients – changes HbF by 7 d/gl (33283989). This proves that editing can make a stronger allele of a GWAS hit than occurs in Nature. It is highly likely that such "BCL11A enhancer"-like targets would be found for common disease (as is the case for CAD and PCSK9 or ANGPTL3).

We thank the referee for these fascinating facts. Our interpretation of this example is that a common GWAS variant with small effect can lead to a gene whose expression can be targeted therapeutically and thereby have a much larger effect on the relevant phenotype. In our polygenic modelling, we suggest that editing of multiple sites with small effects each can cumulatively also lead to a large effect on the trait. We have cited these examples of where gene editing of a single allele can have significant effects (page 7, lines 290-294).

67-78 and 180-181 and 193: the authors could consider citing 32315033 and 36064968 as an

example of multilocus editing in tissue culture cells that has already been attained. The other example to consider citing is the work from Beam 35560156 – it’s a clinically relevant cell and this approach is in the clinic. The “multieditable human genome” future the authors allude to is quite close.

We thank the reviewer for this comment and have now cited these papers in the context of advances in multiple gene editing technologies (page 2, line 86).

93 and elsewhere: the authors should consider mentioning somatic genome editing as a clear alternative. Eg it is important to discuss the ongoing clinical trial by Verve to knock out PCSK9 in the liver (preclinical data 34012082) for familial CAD. Verve has shown strong preclinical data that they can double-edit PCSK9 and ANPTL3 in nonhuman primates. This means that in very practical terms, there will be methods to deliver a CAD-protective genome edit during the postnatal period – without any of the handwringing about embryo PGE. A gene therapy trial has begun to deliver ApoE2 to patients homozygous for ApoE4 (29409358) – and this will assuredly pave the way for editing ApoE4 directly (eg <https://www.businessinsider.com/gene-editing-pioneer-david-liu-developing-drug-to-prevent-alzheimers-2021-11>). It is essential the authors make it clear that such approaches – especially for diseases that are later-onset (eg CAD or AD) – can be ethically advanced through clinical development today, in full compliance with existing laws for protection of human subjects in research, the Belmont report, etc. Thus, by the time – 30 years from now in the authors’ estimate – that embryo editing will technically get to the point to allow safe PGE, there will be a number of approved medicines to reduce risk of common diseases by editing adult individuals who have given informed consent.

We thank the reviewer for this comment and suggestion and have now added a paragraph to discuss somatic gene editing as an alternative (page 4 line 175 – page 5 line 189).

442: further to the point just made, it would be most appropriate to re-cite the recent Doudna review here. CRISPR has been marching through clinical development with trials for sickle, thal, TTR, HAE, LCA, HIV, CAD, cancer – and a number of others to initiate shortly. This is not tangential to the future the authors describe – it’s enabling. The only path to PGE for, say, CAD, is an extended track record for such editing in the postnatal period to treat established disease. When and only when such clinical use is shown to be safe and effective will there come a time to discuss PGE.

We thank the reviewer for this comment, agree, and have re-cited Doudna’s review (page 15, line 540). We now note that one way in which the prospect of PGE will become more realistic is through the increased use and success of somatic cell therapies.

134: the authors’ discussion of using PGE to increase IQ is – in its present form – gravely lacking in key details.

The paper the authors cite identifies 205 genomic loci bearing 1,016 genes as exerting an effect on “IQ” (however one defines that – perhaps “reifies” is a better term) in individuals of European ancestry. The notion that a trait of such complexity can be manipulated by using editing to “introgress 20 smarter alleles” into the genomes of the “unintelligent” is simply untenable. In contrast to diseases such as CAD or neurodegeneration, there is, by definition, no ex vivo animal model where the effects of such introgression could be meaningfully tested in preclinical models – today or ever (unless one is prepared for Jiankui He-level criminal activity on actual human embryos).

Further, and even if one assumes a miracle of a future where there is a brain organoid system where

medium spiny neurons can be assessed for the “IQ-enhancing effects” of this introgression – there is highly unlikely to ever be a delivery modality to the forebrain that would allow comprehensive editing across the entire organ. This means that only embryo editing could ensure that a person with a fully edited brain is born. This reviewer is hopeful the authors agree that attempting such experimentation on embryos for “IQ increase” should forever remain beyond the pale.

Next, there is Mendel’s second law to consider. Imagine a future where – via a set of technical miracles – it becomes feasible to make a human bearing IQ-enhancing variants across 20 loci in homozygous form. Given Mendel 2, after this high-IQ individual has a biological child with an unedited partner, the 20-locus-heterozygous individual resulting from that union will produce gametes with a 20-dimensional Punnett square (less so if some of the loci are tightly linked). How are we proposing to assess the safety of this manipulation (given that it will take 18 years for that F1 to have a child of their own – and we presume at least a decade for that F2 to reach an age where their “IQ” can be assessed)?

Next – what is the plan with respect to validating, specifically for “IQ,” the authors statement (line 234) that human genetic variation is additive and genetic-background-independent? Development of “cognitive ability” is not serum lipid chemistry (where a single-gene KO will be highly penetrant across all genetic backgrounds).

Finally – the paper the authors cite has the following extraordinary statement buried in Materials and Methods: “Different measures of intelligence were assessed in each cohort but were all operationalized to index a common latent g factor underlying multiple dimensions of cognitive functioning.” All of this polysyllabic verbiage can be distilled to one thing: that intelligence, like height, is a single thing that can be measured with one number. Like the authors, this reviewer is not a cognitive scientist. Having said that, debates about reification of “intelligence” to a height- or weight-like quantifiable property have been raging for a century. What is the current consensus on this “common latent factor” as being a meaningful metric of cognitive function? Is it legitimate to assess “different measures of intelligence ... in each cohort” and then act as if they are measuring the same thing?

In sum, the part of Fig 2 (and cognate narrative in the main text) is, in the eye of this reviewer, deserving of significant revision. The popular press will promptly convert Fig 2 in its current form to “Can Editing Genes Make Your Kids Smarter? Yes It Can, and This Will Happen Soon! (say leading scientists writing in the leading journal “Nature”).”. This will happen if this portion of the manuscript remains as-is – surely this is not the authors’ intent.

We thank the reviewer for all the critical comments about the IQ example. In the light of these comments and those by other reviewers we have dropped this example from the paper.

Referee #2 Expertise: statistical genetics, polygenic prediction

In this manuscript, the authors make a set of calculations that indicate theoretical changes in human trait distributions and the prevalence of several major common diseases if multi-variant gene editing of gametes were undertaken at a population scale. These calculations are then used as a grounding for an ethical discussion about the pros and cons of such editing. Considering the importance and sensitivity of the topic, given past (& potentially future) atrocities motivated by eugenics, these calculations and their associated results could hardly bear greater weight. Despite this, the calculations performed here make a range of assumptions known to be incorrect, rendering the results - at best - highly unrealistic, and - at worst - exaggerated and dangerous (given the context). Below I focus on the problems that I have with the calculations rather than the ethical discussion, given my expertise, but reiterate that the ethical discussion stems from, and is thus potentially compromised by, those calculations/results (abstract: "...we use these calculations to ground a discussion..").

We thank the referee for their time spent on reviewing our paper, for making cogent critical comments and for specific suggestions on how to address them. We agree that our benchmarking calculations were based upon many assumptions that are currently not met and are unlikely to be all met in the future. We discussed this briefly in the original submission but, as noted by the reviewer, did not attempt to quantify alternatives. We were initially reluctant to try to quantify any violation from the assumptions because it would necessitate making further assumptions, for example about specific parameters. However, given the substantive comments from the referee we have now performed additional calculations and simulations. Specifically, we have modelled and quantified: 1. the effect of mis-identifying causal variants, 2. off-target deleterious effects, 3. GxE and GxG, 4. pleiotropy with fitness, 5. increase in societal inequality. We have also left out the results on IQ and anthropometric traits and focus solely on disease and risk factors. For the numerical examples we now restrict results to the top-10 loci.

The main problem that I have with the calculations/results is that they rely on a set of highly unrealistic (almost outlandish) assumptions, and while the authors do acknowledge some of these, I think worth stating that the calculations assume: (i) gene editing can be performed at 100 sites without error or off-target effects, (ii) every individual in the population undergoes the same 'polygenic' editing (only possible in a tyrannical dictatorship), (iii) effect sizes derived from genetic association studies are estimated accurately and are causal and are not subject to winner's curse (none of which is true, and unaccounted for winner's curse ensures that all estimates provided are inflated), (iv) there is no pleiotropy (yet we know it is the norm), (v) there are no G*G interactions (i.e. effect sizes are additive across multiple variants even when additivity implies increases in height of 8 SD units and of IQ of 5 SD units), (vi) no G*E interactions (e.g. no 'feedback loop' effects after dramatic changes in depression prevalence or population IQ), (vii) there is no action of natural selection (e.g. limiting such dramatic phenotypic effects; yet recent work, including by the 1st author, has shown that past/ongoing selection on complex traits has been pervasive and critical to observed polygenic architecture).

(i) We have now modelled the effect causal variant misspecification (Supplementary Figure 2) and off-target effect (Supplementary Figure 1). They show that actual change can be much less than predicted and that fitness can be reduced.

(ii) We did not intend to present our results as if the entire population undergoes the same genome editing and specifically labelled the results (e.g., Figure 1) as changes 'among edited genomes'. We have now clarified this more strongly and have also performed calculations about a

possible increase in inequality (in disease risk) as a function of the proportion of the population that undergoes genome editing (Figure 3).

(iii) We thank the reviewer about the comment on winner's curse and agree that any over-estimation of the effect size will lead to an outcome that is less than that predicted by genome-wide significant loci and cite a relevant paper. We now specifically mention winner's curse as one of the reasons why actual outcome may be less than that predicted (page 6 line 245 – page 7 line 249).

It is common in statistical genetics studies to simulate a range of scenarios that are relatively realistic given what we know, and I see no reason not to have done that here. Results based on a range of more realistic scenarios (e.g. 1% population uptake of editing, some antagonistic pleiotropy, stabilising selection reflecting that observed in real data) could have grounded a more nuanced, relevant and urgent ethical discussion, involving issues such as: the impact on population differentiation (& society) if only wealthy individuals and countries have high uptake of editing (as is **extremely likely**, rather than merely one of two possibilities as described by the authors), how to decide which diseases should take precedence when there is antagonistic pleiotropy between diseases, is a (realistic) predicted reduction in disease prevalence worth the potential reduction in flexibility to adapt to future environmental changes (e.g. infections), and are the potential gains of polygenic editing over single/oligo editing worth the potential risks. It is a pity that the results presented are too unrealistic and cursory to inform such (needed) discussions and instead provide the opportunity for broad, and potentially dangerous*, misinterpretation. I would have expected the presented results to only be shown (e.g. in Supplementary Material) as theoretical (but implausible) upper bound results to show predictions in the absence of real processes (pleiotropy, selection etc).

We have tried to address these questions and comments by modelling (new Figures 1 & 2 and new Supplementary Figures 1-3) and by expanding the discussion sections. These results show that outcomes may be much less than predicted by misspecification of causal variants or changes in future environments, that fitness can be comprised through off-target editing and/or through large phenotypic changes under stabilising selection and that societal inequality can increase depending on the proportion of the population with edited genomes.

(*why wouldn't one of the world's numerous dictators see these results and associated ethical defences as an instruction manual for making their population disease-free, 2-feet taller and 75 IQ points smarter than all other populations, within one generation? In practice, attempted implementation of such widespread gene editing could lead to population wipeout, but the reasons for that are not highlighted in this article, while the (likely highly inflated) 'motivation' is)

Given potential public reaction, I would personally recommend only seeking to publish this manuscript (in any journal) once the authors have performed comprehensive, realistic calculations that explore plausible future scenarios (10-15yrs from now.. which gives enough time to have ethical discussions grounded in relation to those predictions and time to update those calculations in ~5yrs time based on newer information, such as editing quality and uptake rates), alongside corresponding ethical discussion. However, in case useful, below are more specific issues that I have with the calculations in particular:

Almost no information is provided about the genetic variants theoretically edited in relation to the figures (apart from e.g. two variants described in Supplementary Box 1). It would be useful to at least provide details of the first 5 or 10 genetic variants prioritised for editing for each of the traits/diseases, giving details of their allele frequencies, effect sizes and the sources of those estimates (e.g. which GWAS/paper).

We thank the referee for this suggestion. We cited the relevant papers from which we derived the GWS loci, provide lists of the those loci (in Code Availability) and now list the top loci underlying Figure 1 in Supplementary Table 1. We have also greatly expanded the theory section and give worked examples for a disease and quantitative trait.

I found Supplementary Box 1 really useful for quickly checking how the authors worked out the impact of editing one variant on disease prevalence, but I think it would be useful for readers to run through the calculation relating to the first two variants edited for several of the traits/diseases (which they can then match to the figures) because once the reader knows how that is done for two variants then they can extend for any number of variants.

We thank the referee for this suggestion. The general calculations for both quantitative traits and disease were given in the Supplementary Information which we hope the reviewer had access to. We now give a specific example for LDL and Type 2 Diabetes for the first two loci (Supplementary Methods) and have expanded the theory sections.

From reading the article one could believe that it is the ‘polygenic’ editing of many variants that has caused such dramatic changes in traits/diseases presented here - but really much of those changes are the result of editing one or just a few variants for which there is a rare protective allele. If this were clear, then it would also be clear that the ethical discussion here is essentially grounded on terms similar to that of the many ethical discussions published since the editing performed by He Jiankui of a HIV-resistance variant (i.e. that editing of individual rare protective alleles can dramatically modify individual/population risk).

We fully agree that if there are rare protective alleles then those will most strongly affect the predicted changes and thank the referee for the suggestion to focus on fewer loci. Indeed, for a number of the traits for which we performed calculations the changes are driven by a few rare protective variants. We now give the results from Figure 1 for the first 10 loci, which shows that for Type 2 Diabetes, LDL cholesterol and triglycerides, the results are driven by a small number of (protective) variants. The numerical examples in the Supplementary Methods also highlight that the top-2 loci for LDL and T2D have risk variants are very high frequency, so rare protective variants.

Interpretation would be improved if the X-axis included 0 edited alleles (trivial for traits, but especially for disease prevalence this would make clear the change that just one allele is edited Vs subsequent edits). I don’t think it is appropriate to display prevalence (Y-axis) changes on a log-scale, which exaggerates the impact of editing many, compared to a few, alleles. Making these proposed changes to the axes would help to address the point made in the previous paragraph.

We have added 0 to the x-axis and have parameterised the predicted change in disease as a fold-reduction in prevalence.

The results presented are population-level (changes in trait distribution / in disease prevalence) and yet at various points throughout the ms the authors talk as though they have made individual-level inferences, e.g. in the abstract “Editing a relatively small number of variants can make a substantial difference to an individual’s risk of developing coronary artery disease, Alzheimer’s disease.. “. Clearly if an individual is homozygous for the APOE-e2 allele then the proposed edits will not lead to a reduction in disease risk corresponding to the reported reduction in disease prevalence. The variance in individual changes in traits and disease risks has not been considered here whatsoever, but would be interesting to estimate and present and would allow comments in relation to

individual-risk.

We thank the reviewer for this comment. We did not consider the variance because if the number of loci is large enough then the variance is small relative to the predicted change. We have now quantified the variance by calculating predicted change among multi-locus genotypes, conditional on allele frequencies and effect sizes, and have added the standard deviation in outcomes in Figure 1. We have also expanded the Methods section to quantify this variability theoretically.

The authors state in the abstract “Our modelling shows quantitatively how putatively positive consequences of gene editing at an individual level may deepen existing social inequalities”. This seems an unusual statement given that no part of the modelling performed considers potential impacts on social inequalities. I think it would have been really interesting to investigate this, but given that no such modelling was performed, the article effectively leaves the impact on social inequalities open to widespread/public interpretation, which I think is risky given the potential stakes here.

We thank the reviewer for this comment. We have now attempted to quantify inequality by calculating a “Gini index” for the probability of disease for a hypothetical common disease (Figure 3 and Supplementary Methods).

Stabilising selection on human traits has helped to shape observed genetic variation and the genetic architecture of traits, presumably including selection against certain high-order combinations of alleles present in the population. In fact, this is nicely demonstrated by the 1st author of the present manuscript in Sanjak et al 2017 (PNAS, doi: 10.1073/pnas.1707227114), in which it is stated “for several traits, we demonstrate that individuals at either extreme of the phenotypic range have reduced fitness”, one of which is height (see Fig.2). Thus, I find it surprising that the 1st author considers it reasonable to present results that ignores the likely impact of such selection, inferring that genetic editing of multiple alleles could result in a 8 SD increase in height in the population (even without considering the implicit physiological birth-related issues). The results from Sanjak et al 2017 (see Fig.2) could in fact be used to make the case that even a small number of height allele edits could reduce fitness/fecundity and thus the population size due to selection (or at least in those undergoing editing).

We agree and show new results in Supplementary Figure 3. It shows that a substantial reduction in fitness can result when large phenotypic changes in a trait are achieved through genome-editing when the current population is at an optimum.

The fact that the calculations predict almost complete eradication of depression in the population on editing 50-100 genetic variants seems to highlight the disconnect between the calculations/results and the likely interplay between genetics and the environment (as well as other limitations of the calculations discussed above). Even if baseline genetic risk for depression is modified dramatically in all individuals, factors such as societal position, interactions among people, competition etc, would surely result in depression at substantially higher rates than predicted by assuming that modification of genetic variants can impact risk in isolation from the environment (/resulting environment).

We agree with the referee that environmental effects and genotype-environment interactions on the scale of liability would lead to actual changes in edited genomes that are less than that predicted, and that disorders for which prevalence has changed over time (such as depression and T2D) are good examples of that. We have tried to model general GxE effects (Figure 2) and have

added text to emphasize that changes in environment and/or interactions between genomes and environments will lead to reduction in phenotypic change compared to the prediction.

It would have been nice to see some investigation of real data, to test for enrichment or deficits of individuals with combinations of the alleles proposed for editing here (indicating viability / pleiotropic effects / selective pressures).

We understand this suggestion to mean that we correlate fitness with a polygenic risk score calculated from the loci we 'propose' to edit for the traits considered. We reported significant (linear) genetic correlations between fitness (relative lifetime reproductive success) and traits in Sanjak et al. 2016 but did not attempt a quadratic regression because of lack of power (and provided power calculations). In our opinion, a thorough analysis of stabilizing selection at the genetic level of the example traits considered would require more data and is beyond the scope of the study. However, we have given results for a model of stabilising selection (Supplementary Figure 3).

“Human enhancement” is rather a loaded phrase and is subjective - an explicit assumption is made here that increases in height/IQ corresponds to human enhancement, but I can think of a few arguments against this claim. Also, the controversy around the meaning of IQ and the genetics of IQ are ignored here - I cannot speak to the former, but in terms of the latter it has been clearly demonstrated that much of the previously estimated heritability of educational attainment is due to the ‘household effect’ (i.e. is due to the environment created by parents), and I suspect similarly for IQ, while some of the remaining heritability will be explained by G*E correlation, and so the identification of G-only causal effects on IQ may be particularly challenging.

We used the word ‘enhancement’ because it is a standard term using in the discussion of human heritable genome editing (as, for example, in a number of presentations at the March 2023 Third International Summit on Human Genome Editing, <https://royalsociety.org/science-events-and-lectures/2023/03/2023-human-genome-editing-summit/>). We have now dropped IQ, BMI and height from the paper.

While the authors claim that PGE could have a far larger impact than ESPS is true in theory, I think the opposite will likely be true in practice. ESPS is presently legal in many jurisdictions and can become substantially more efficacious with relatively minimal technological advance (generation of many more embryos).

We thank the reviewer for this comment. We agree that ESPS is likely to be used in practice sooner (and already is in some jurisdictions). ESPS is a form of within-family selection which is not particularly powerful (in particular when selection is on multiple traits that are unfavourably genetically correlated) and relies on variants segregating in a family, so common variants and not for example rare protective variants. Larger changes through ESPS could be achieved if selection could be performed on 10,000s of embryos per couple, for example using in vitro gametogenesis, but that is not currently possible and unlikely to be socially acceptable (Bourne et al. 2012; Karavani et al. 2019).

We have added the following paragraph in the Introduction:

“ It is currently not possible to use embryo selection on PGS to achieve large scale changes in polygenic conditions (Karavani 2019; Turley 2021). Theoretical calculations imply that 10,000s of embryos would be needed per couple to achieve a one standard deviation change in phenotype (Karavani 2019). Creating such large numbers of embryos may be possible in the future through in

vitro gametogenesis but is not feasible at present and unlikely to be socially acceptable (Bourne 2012)."

Referee #3 Expertise: population and evolutionary genetics

Visser, et al. have written a prospective piece on human germline genome editing that contains a good amount of nuance but ultimately comes out in forceful support for the idea that we should eventually consider editing embryos with the aim of altering complex traits. If published, the piece is sure to be extremely controversial—I'd guess it will dwarf the controversy that surrounded the publication of Harden's Genetic Lottery (a text that should probably be cited when the authors reference a "genetic lottery" in line 355).

We thank the referee for their time spent on reviewing our paper, for making cogent critical comments and for specific suggestions on how to address them. Our intention is not to create controversy but to contribute to, in our view, a much-needed informed discussion about the potential uses of human heritable gene editing for complex traits. We have now cited Harden's book (page 13, line 433). We note the statement from the Third International Summit on Human Genome Editing (<https://royalsociety.org/science-events-and-lectures/2023/03/2023-human-genome-editing-summit/>): the organisers concluded that safety and efficacy for heritable human genome editing has not been established, nor has the societal discussion and policy debate been completed. They also stated that human germline genome editing for research purposes should continue. Our paper is meant to span these statements, by performing research to aid societal discussion.

The first section of the perspective invites us to consider a world in which we are capable of precisely editing a human embryo at tens of loci, and in which we have fine-mapped human complex traits successfully enough to know where to make edits to most effectively perturb traits like diabetes risk, height, and IQ. Within the context of this model, the authors calculate how much we might perturb various human phenotype by means of editing a particular number of loci. The second section of the paper then explores ethical pros and cons that should be considered before deciding whether to apply such a hypothetical gene editing technology at scale.

We thank the referee for this summary.

The concrete predictions that are made in the first section of the paper are bolstered with some technical supplementary information, but I did not find this technical supplement to be detailed enough to judge the correctness of the numerical claims made in the text and in Figure 1. The supplementary note is clearly written, but it is extremely succinct, and it omits definitions of many necessary technical terms (including f_0 , λ , K/K_g , ϕ , and others). These omissions aside, I think that this mathematical framework seems basically sound in that if you input variant frequencies and effect sizes, you can accurately calculate the effects on the population trait distribution that will result from editing these loci. However, the paper includes no details about what concrete frequencies and effect sizes they plugged into this model to obtain the predictions in Figure 1, which claims that editing tens of loci should be enough to change various trait means by multiple standard deviations. These effect sizes seem extremely large in light of the modest amount of trait heritability that is explained by published GWAS, and should not be published without a more detailed accounting of the parameters that were used to derive these results.

We thank the referee for this comment. We have now expanded the theory section and give specific numerical examples underlying Figure 1 (Supplementary Methods). We also provide a Supplementary Table with details of the top loci. We agree that GWAS effect sizes are typically very small and that each locus explains a tiny proportion of phenotypic variation in the population. However, the effect of gene editing is on the mean of the trait and not its variation. For example, a hypothetical variant with a frequency of 1/1000 in the population that decreases diastolic blood

pressure by 7.5 mmHg explains very little variation (about 0.1%, assuming Hardy-Weinberg Equilibrium and a population standard deviation of 9.8 mmHg), and 99.8% of the population are homozygous for the ‘increasing’ allele. Yet editing that variant to become homozygous for the ‘decreasing’ allele is predicted to decrease blood pressure among the edited genomes by 15 mmHg, a large amount.

The authors do cite particular genome-wide association studies that they presumably used to obtain the allele frequencies and effect sizes that yielded their predictions, but several of the studies they cite are affected by problems that the authors do not address in the paper at all. For example, they cite GIANT consortium association studies for height and BMI, but do not mention a pair of papers (Berg, et al. eLife 2019 and Sohail, et al. eLife 2019) that showed that the GIANT consortium likely severely overestimated these trait associations due to population stratification. This stratification problem also affects the conclusions of Field, et al. 2016, which is cited in support of the idea that these traits are rapidly evolving. If the authors used beta values from the GIANT papers in their model, they are likely overestimating the impact that editing a few loci would have on height and BMI. It is also likely that GWAS of traits other than height and BMI are subject to as much or more confounding by population stratification, meaning that all of the effect sizes output by the authors’ model are likely to be overestimates. I don’t expect the authors to correct for this systemic effect, but it should be mentioned so that readers are aware that these variant effects may be overestimates.

We agree and have now added a paragraph to emphasize that any over-estimation of effect size, whether due to population stratification, winner’s curse or other factors, will lead to an over-estimate of the effect of gene editing (lines 245-249, page 6-7). We also omit height from the revised paper.

Even if we take the results of the GIANT consortium at face value, I am still puzzled by the paper’s assertion that editing ~20 BMI-associated alleles is enough to reduce BMI by a full standard deviation, given that the GIANT consortium found that 941 genome-wide significant SNPs only explained 6% of the total population BMI variance. The authors make no mention of releasing code or the table of data that went into Figure 1—without such data, it’s not possible to judge whether these results are reasonable, especially when the magnitude of the results seems so high compared to estimates of trait variance explained in the cited studies.

We thank the reviewer for the question and suggestion. The large predicted effect on BMI is for the same reason as the toy example above for blood pressure. We now give the code underlying the figures and also provide a numerical example for LDL and T2D (Supplementary Methods). We have omitted BMI and height from the revised paper.

One stylistic choice that I found extremely problematic was the discussion of editing height-associated and BMI-associated loci to achieve unidirectional “desirable” changes in these traits. Although the authors acknowledge that extending trait values beyond their natural ranges could cause problems, they still seem to take for granted that everyone would prefer their children to be as tall and thin as possible, which is frankly bizarre and stigmatizing to individuals and populations that are shorter and heavier than others. The discussion of these traits does not even square with the paper’s thesis that gene editing should be used to reduce morbidity and mortality, given that height has no effect on mortality and that “overweight” individuals actually have lower all cause mortality than “normal weight” individuals (Flegal, et al. 2005). In line 203, the authors refer to high BMI as a causal risk factor for various diseases, but do not note that this causality is debated—for example, much of the decreased disease risk that is sometimes attributed to weight loss better attributed to diet and exercise changes that sometimes but not always cause weight loss (e.g. Ross

and Janiszewski 2008). This confounding takes away credence from the authors' assertion that genetic modifications to BMI should cause the same disease risk reductions that environmental weight loss interventions produce.

We thank the referee for the comment about the discussion about unidirectional changes for height and BMI. They were meant to benchmark what changes could be achieved in theory and not meant to indicate that such changes were desirable. We have removed BMI and height as example traits from the paper.

The assumption that higher intelligence is always more desirable is perhaps less eccentric than the papers' assumptions about the desirability of height and thinness, but it is still stigmatizing to individuals with lower IQs. In addition the statement about selection on loci associated with educational attainment (line 646) is a bit misleading, since most of this paper takes for granted that more intelligence/educational attainment is better while the cited sources claim that there has been selection *against* variants associated with increased educational attainment.

We have dropped IQ altogether and clarified the statement about selection (lines 873-874). We give the examples of polygenic adaptation and selection at loci that are associated with educational attainment to show that genetic change already happens in a short timeframe, irrespective of making a judgement about the desirability of the direction.

The paper could make all the points it is trying to make without perpetuating as much stigma if it restricted its scope to the idea of reducing disease incidence without taking for granted Western culture's assumptions about which anthropometric traits are "desirable" and "undesirable." It would not be a bad idea to expunge "desirable" and "undesirable" from the paper's lexicon.

We thank the referee for this comment. We have changed the text to reflect that unidirectional changes for any trait can be culture dependent (page 11, lines 362-363) and have removed the height and BMI examples.

The ethics-focused portion of the article does a reasonable job of articulating the main debates around the advisability of germline genome editing, but I thought the ethical implications of off-target editing errors deserved a bit more discussion. I realize that the authors are talking about a hypothetical future in which editing error rates are extremely low, but I don't think it's reasonable to assume that editing will ever be error-free. This means that for editing to be acceptable, we need to decide that the benefits of editing outweigh the risks of harming the embryo. Would it every be acceptable to change a cosmetic characteristic like height at risk of introducing mutations that are harmful to health? A similar calculus needs to be applied to the discussion of pleiotropy and the uncertainty of genotype-phenotype associations—one could argue that editing a genome to introduce a rare variant is never ethical because the rare variant has not been tested on enough genetic backgrounds and might introduce a deleterious effect that is not known from genome-wide association studies. I don't agree with the assertion in line 235 that experiments have shown most genetic effects to be additive—I think a more correct assertion would be that when alleles are rare or effect sizes are small, an additive model works to a first approximation, but this would not apply if we used editing to suddenly make a rare allele of large effect into a common variant (as in the discussion of using rare "protective" variants to lower cholesterol). In addition, although the table of arguments for an against PGE is reasonably useful and comprehensive, it contains some sound bites that are pretty unsavory, e.g. "not all forms of diversity are valuable" and "human bodies have clear design flaws." Who gets to decide what kinds of diversity have enough value to be allowed to exist, and classify a feature of someone's body as a "flaw"?

We agree and have modified our language accordingly.

I found the discussion of eugenics reasonably complete and balanced, but it also comes rather late in the article given that eugenics will be many readers' first mental association with germline genome editing. It could be a strategic choice to move much of this material into the introduction of the paper. On a related note, in line 431 the authors cite the idea that the human gene pool is accumulating increased genetic load due to modern medical care, and if they cite this idea, they should acknowledge that it is highly controversial (e.g. see Roth and Wakeley 2016). I would also take issue with the statement in line 455 that we have an "obligation to to manage genetic risks [sic] factors"—the idea of "managing" the human gene pool in any kind of top-down way is a eugenic idea that compromises the ultimate freedom of individuals to decide whether and how to reproduce. I would urge the authors to at most recommend that some forms of PGE be made available for individuals who wish to use it and refrain from the implication that anyone should ever be "obligated" to make use of such technology.

We thank the reviewer for these comments. We have now mentioned 'eugenics' in the Abstract (line 25) and Introduction (line 101). We have also cited both Roth & Wakeley 2016, Lynch's response and Teicher's 2018 historical perspective, and state that the concept of 'mutational load' is controversial (page 14, lines 509-513). We have removed the statement about managing the gene pool.

Referee #4 Expertise: ELSI (biomedical technologies)

I think this is a useful piece. We've been talking and writing about these issues for over 50 years but looming reality, like the prospect of hanging, has a wonderful way of concentrating the mind. (Though you might want to note a bit more the age of the debate on genetics and throw in a few more cites to the pioneers of the discussion.) Your piece should force people to take these issues more seriously and to focus their attention on important "details," like possible regulatory schemes.

I do have one major reservation about it as well as a number of smaller comments.

We thank the referee for their time spent on reviewing our paper and for their positive and constructive comments. We now note that this debate has been ongoing in bioethics since the 1970's and cite some of the pioneering work (page 8, lines 321-323).

My major reservation is that your conclusions about the size of the effects are so startling as to border on unbelievable. Increases of 75 IQ points or 50 centimeters in height (sports would certainly be different if the tallest people were 9 feet tall with adjustments needed to basketball nets, soccer goals, and volleyball nets among other things!) seem fantastic.

We thank the reviewer for this comment. Given the feedback from the Editor and other reviewers, we have decided to remove the examples of IQ, BMI and weight. We agree that the results for the quantitative traits look unrealistic and that for the reasons we discuss actual changes may be (much) less than predicted. We note that in other species (in model organisms and agriculture), huge changes have been achieved over time by selection on polygenic traits, through small changes in allele frequencies at many loci. Those changes have been gradual, and it is unlikely that large changes (as those cited by the reviewer) would be observed by gene editing in one generation, at least not without risk of serious negative side effects. We now model such effects directly by assuming a stabilizing selection model (Supplementary Figure 3).

I do not have any expertise that allows me to understand, let alone criticize your model. I do hope the editors have sent this paper to some experts who can. If such reviews lead the editors to conclude that the projected magnitudes of the changes you foresee are outlandish—not just in terms of issues with your particular model but from limitations in the power of PRS—then I think this should not be published. If, as I rather suspect, such reviewers would say something like "these estimates seem very much on the high side but it is plausible that some very large effects might (or might not) follow." In that case, I would recommend publication.

That's out of my hands (and expertise). But I do think that if that verdict allows the paper to go forward, it still should have even more caveats about the size of the impacts. You have some, but should have more, perhaps along the lines of pointing out although your estimates might be at the high end of plausible projections, that even if the increases are "only" half as large, they are very significant. Your paper's impact comes from the idea that this technique could make big changes but even if someone came away from the paper thinking there is only a 20% chance you are right, or even close to right, about the scope, that should still motivate increased practical attention to these issues.

We thank the reviewer for these comments. In response to comments from the Editor and other reviewers we have now expanded the modelling section to quantify how actual changes could be less than that predicted under simplified assumptions, and where they may be detrimental (pages 4-6 and Supplementary Information, page 20-22).

Now for some other comments, ranging randomly from small to large.

You cite the George Daley et al NEJM paper on paths forward. I think you should, in the same vein, cite the reports from the WHO and NASEM/Royal Society working group/commission.

We now cite the 2020 report from the International Commission on Germline Genome Editing (NASEM and Royal Society), the 2021 WHO report and the 2023 Third Summit statement (page 1, lines 42-43).

Make it clearer that Aurea was selected, not edited. You say that initially but by the end of the paragraph, I worry that that nuance will be forgotten. In general, I think the piece should emphasize even more how PGE is different from PRS PGD.

We thank the reviewer for this comment and have now expanded the text to emphasize how selection is different from editing, and specify that Aurea was selected (page 2, line 50).

I do think you are very much on the bullish side on the power of PRS. And the fact that you are citing a 2018 paper for its big claims that PRS can (often be?) as powerful in its predictions as those for monogenic conditions actually makes me nervous. That's five years old but the controversy in the field is (as far as I can tell) still raging. (I do like the point about rare protective alleles, where I can see a strong effect as quite plausible.)

We thank the reviewer for this comment. To our knowledge there is no controversy about the predictive power of PRS because they have been validated repeatedly in out-of-sample prediction. Since 2018 there have been a number of traits where the GWAS sample size has increased substantially (in particular the 'model' trait human height), with increasing prediction accuracies in out-of-sample prediction (e.g., Yengo et al. 2022, Nature). We cite Yengo 2022 in the Introduction (line 67). In that paper, the "effect size" of a PRS (effect per standard deviation) is demonstrated in out-of-sample prediction, in populations of different genetic ancestries.

When you get to the limitations in the model (which, to be fair, seem to me to be major limitations in all the models showing strong effects for PRS, not just yours, I think you should change their order. The two most important limitations are the ones you list fourth and fifth: epistasis and environment, followed closely by pleiotropy. The first two are largely technical (although I do suspect that more samples leading to more "causal" variations will also increase the significance of the problems of epistasis and environment. I'd take them in order of hardest to easiest, in the spirit of putting the biggest risks first in the consent form. (By the way, I'm not sure epistasis exactly is behind my own suspicion that models make too many assumptions about how the various polymorphisms interact, often by assuming they are additive when there is no particularly reason for them to be. But I can't even tell if your model is an additive one.)

We thank the reviewer for this comment and suggestion. In the short term, the first two are the most important because they affect feasibility today and environments are unlikely to change rapidly in the short term. But in the longer term, if these technical issues can be resolved, the environment and pleiotropy seem the most important to us, because there is lack of evidence for epistatic effects on complex disease variants. We have changed the order and after listing the technical challenges now list the environment first (page 4-5, lines 172-189).

I didn't like "Other countries, notably the US, adopted "negative eugenics". Lots of countries did so, including Canada and much of Northern Europe, especially, of course, Nazi Germany. Later you

include some more countries, but it seems to me unfair to single out the US in your first mention on negative eugenics.

We have changed this sentence (page 8, line 302).

I don't disagree with your statement that "previous eugenics practices...were clearly unethical", but you might want to give a reason that at least forced sterilization was clearly unethical, presumably the coercion involved.

We agree and have changed the wording accordingly. We explain that early eugenics practices of the 20th century were clearly immoral not only because of the coercive practices used, but because they were directly harmful, relied on false scientific views about heredity and thus had no prospect for benefit. While it is important to keep in mind the lessons of early eugenics programs, they can be clearly distinguished for use of genetic technologies to reduce the incidence of disease, in ways compatible with human rights. Page 8, lines 302-204: "Many other countries adopted "negative eugenics" policies, which imposed severe, unethical restrictions on peoples' individual liberties in order to prevent those with 'undesirable' genes from reproducing, included policies of forced sterilisation (ref 37)."

I'd add to the response argument for enhancement that we already are very different from past generations, obviously in cultural ways but also in biological ones.

The "non-identity argument" in box will puzzle non-specialists. I think you need to explain it more clearly (and good luck doing so in only a few words).

You talk about the expressivist argument and its effect on those with disabilities and both the argument and the response are fair, but I would also add that the elimination or great reduction in disease incidence will have more concrete effects on people with disabilities—less research, fewer specialists, less social support. (Personally, I think the disability issues are the hardest, for me, in genetics.)

You talk about surveys of "people in most regions" but how many regions? How many of them were sampled? And how fully? I suspect most of the world's cultures and peoples were greatly underrepresented, but maybe not. Some specifics would help.

The argument on the difficulty of drawing the line between enhancement and treatment has never seemed strong to me. Yes, there will be fuzzy zones. No, the vast majority of applications will not fall into those zones. And we have ways of drawing, imperfectly but usefully, lines in fuzzy zones. argument isn't very strong

We thank the reviewer for these suggestions. We now note that human populations have changed dramatically in the last few generations via cultural means (page 10, lines 336-338). However, the prospect of radical, fast changes to human physiology is thought to raise unique ethical concerns.

Re the non-identity problem. We have added to the box in Table 1 such that it states "the non-identity problem is a puzzle in ethics when our actions today might change who gets born in the future. It can be hard to specify how these actions can be harmful to people who would not have existed otherwise, unless they have lives which are not worth living."

Re the reduced resources objection, we now note this problem (Table 1). We agree that approaches to disability are among the hardest ethical issues for germline gene editing. However, in this case, scenarios where reductions in the incidence of a disability have no effect on the resources available to the remaining individuals with that disability, or indeed increase the resources available to them, are conceivable. As a real-world example, when carrier screening was introduced in Cyprus for beta-thalassemia, it reduced the incidence of the disease, but the resources available funding for beta-thalassemia support programs remained constant, resulting in more resources available for each of the remaining sufferers. This shows that a negative effect on resource availability is not an inevitable consequences of germline gene editing to reduce rates of disability.

We have modified the Expressivist Objection in Table 1 to include this objection and provided a response to it.

As we have now dropped the examples of IQ, height and BMI, the issue of enhancement is less pertinent to the paper. We have rephrased this section and dropped talk of the fuzzy zones between therapy and enhancement.

We have also considered the treatment-enhancement distinction and added the possibility of a welfarist approach (page 11, lines 362-366): “One approach to the use of ESPS has been to limit its use to selection of traits with a reliable relationship to a robust conception of well-being (Munday 2021). This of course raises the vexed issue of which conception of well-being to employ, but this is a problem for any welfarist approach to societal improvement.”

I don't like your AD example. I suspect simple multiplicative combinations won't work for it because the odds of having each of those ten won't be independent of each other (even if such alleles are real and work in something like an additive manner, as to both which I would need to be convinced).

We thank the reviewer for the comment about AD. To our knowledge, the models that best explain the observed data for common disease (probability of disease as a function of the effect at multiple genetic loci) are those that are multiplicative on the observed prevalence scale and additive on the log-risk (or log-odds or liability) scale (e.g., Pang et al. 2022 “Genetic and modifiable risk factors combine multiplicatively in common disease”, *Clinical Research in Cardiology*). We have tried to clarify the choice of models in the paper and have cited the Pang paper in the Supplementary Methods section (page 20, lines 736-737).

I'm also troubled by your “populations may benefit sometimes from some prevalence of a harmful allele” argument, not because I don't believe it but because if we learn THAT much about the genetics of common disease, we'll be able to treat it even (especially?) in people at high genetic risk. Medicine is part of the relevant environment, and it should advance at least as much as our understanding of the meaning of GWAS and PRS.

We agree and the revised manuscript does not contain the claim that populations benefit from the presence of harmful alleles.

By the way, is there any reason to think that there are any alleles with solely positive effects? And if there were, how in the world would we know that—proving a negative is going to be hard, especially if that negative means showing it might not be harmful in the face of a new pathogen or new environmental factor.

While the effects of all alleles are context dependent, we do think here are some alleles which have robustly positive impacts on organisms - in the sense that any variation in these alleles would be harmful. For example, the alleles that encode the components of the Krebs cycle are highly specialised, and variation in them likely lethal. However, we take the reviewer's point and have qualified our language throughout. We have also modelled scenarios where deviations from the current population mean by large amounts leads to deleterious effects on fitness (Supplementary Figure 3).

I really hate the “genome degrading” argument, and not just because it is so reminiscent of the early 20th century eugenicists (“The Death of the Great Race, etc.). As you know, and note, the significance of genetic variations (and phenotypic ones for that matter) depend on the environment. Insulin undoubtedly led to a higher percentage of people having risk alleles for type 1 diabetes because now they could survive to reproduce. So what? If we can treat type 1 diabetes (or, to be fair, if we could treat it perfectly, which we can't now but should be able to do before you can do PRS editing), why does that make the genome degraded? We have undoubtedly lost (or had lowered incidence of) some alleles good for being a hunter/gatherer. So what? And the word “degeneration” is just creepy. You should eliminate it.

We thank the referee for this comment, which echoes that of reviewer #3. We agree with the need to be careful with the language here. The word degeneration has implications we wish to avoid, and we have followed the reviewer's suggestion to eliminate it. We have now qualified this section of the paper by stating that the mutation accumulation argument is controversial and cite Roth and Wakeley (2016) and Teicher (2018) (page 14, lines 510-513).

The issue with relying on treatments for diseases like diabetes, rather than genetic prevention, is the opportunity cost. Developed countries spend huge portions of their health budgets on treatments for polygenic diseases, which could be deployed elsewhere if the incidence of these diseases were much lower. This may put future generations under greater risks from other challenges. For example, if nearly all adults need access to diabetes medications in the future, the effect of natural disasters which disrupt the supply of health resources will be much greater. It is an open question whether treatment or genetic prevention is safest and most cost-effective in the long term.

Having said that, I do like (mainly) your argument for a more collectivist approach, although spelling that out is probably more the work of more than one book.

We thank the reviewer for supporting this approach. We agree that further elaboration on how collective approach to genome gene editing is an important task going forward for debates about germline genome editing.

I wish you would spare a thought, and a paragraph, for the practical ethics of starting down this path. Who is the first PGE'd baby? How safe and effective does it have to be before we are willing to risk a baby's life and health on it? And how would we know how safe and effective it was? One good reason for dealing with awful diseases is that the risks of non-intervention are often great enough to justify some unknown risks of intervention. PRS will normally involve smaller benefits, especially if medicine continues to progress.

One last idea – when discussing massive editing, you might throw in a sentence about genome or chromosome synthesis. In 20 or 30 years, it may be cheaper and easier than making 2000 edits.

We thank the reviewer for this comment which is similar to a comment from reviewer #1. We now explicitly discuss that somatic genome editing or other somatic technologies might obviate the need for heritable genome editing (page 4, lines 173-177). We have now added a new paragraph about a possible translational pathway to polygenic gene editing (page 14, lines 520-537).

So, overall I've got a lot of comments that I hope you will consider, but few that I think really must be incorporated in a revision. I enjoyed reading the piece (except for the model, which I didn't even attempt to read.)

We thank the reviewer again for the positive and constructive feedback.

Referee #2 (Remarks to the Author):

The efforts that the authors have made to incorporate some more realistic modelling scenarios have helped to ground a more informed ethical discussion. However, I think the manuscript still has a number of issues that should be addressed, especially given the likely attention and possible consequences of the paper.

We again thank the referee for their time spent on reviewing our fully revised paper and for acknowledging our efforts to improve upon the initial submission. We have made further changes to address the reviewer' comments, made textual changes to improve clarity and changed the main acronym to HPE (heritable polygenic editing) to avoid confusion with polygenic prediction acronyms and pre-implantation testing acronyms.

Evaluating the paper as it is now, I think the statistical genetic modelling demonstrates two key points that - despite being simple and intuitively clear even with little/no modelling - most people in the field and beyond may not have considered:

(1) the editing of a small number of alleles (< 10) could have a dramatic impact on individual risk of major complex, polygenic diseases and the values of complex traits.

(2) as more alleles are edited for each disease there are diminishing 'gains' and increasing risks (off-target effects, antagonistic pleiotropy, severe effects of stabilising selection, reduced diversity), which at scale in a population could lead to catastrophic consequences (large-scale death, prevalent unviable foetus', major population reduction or wipeout).

I think these points could have dramatic consequences for the future of medicine and society and do represent the grounding for an important ethical discussion on the topic that could be valuable to initiate now. However, I think the ethical discussion should be framed more in relation to the spectrum that these two points highlight - i.e. that editing a small number of alleles has the potential to reduce risk of major disease(s) in edited individuals with incurred risk that may be outweighed by the benefits (analogous to medication and its side effects), but that editing a large number of alleles is likely to produce risks to the individual and population that outweigh the benefits [Note: my comment here relates only to statistical and population genetics inference, not to any ethical issues]. Or to put it another way, "oligogenic editing" of genomes (< ~10 edits) may have clear health benefits to individuals that society should at least consider seriously (inc. the ethical implications) given the goal of healthier populations, while "polygenic editing" of genomes presents likely catastrophic consequences for both individuals and populations (of the scale of the potential consequences of climate change, nuclear power and unregulated AI). While there's no obvious line between oligogenic and polygenic, I think it's clear that any predicted benefits from editing become serious dangers somewhere along that continuum.

The authors do highlight that "editing a relatively small number of genomic variants" (abstract) can reduce disease risk substantially, and all their main figures now relate to 10 or fewer allele edits, but throughout the ms the authors talk about the effects and ethical implications of "polygenic editing" (including in the title). This conflation causes unnecessary problems, because polygenic editing should be able to be ruled out as a clear danger by the authors (barring unforeseeable changes) on the basis of present knowledge and their own modelling, whereas the real modelling and ethical uncertainty is whether oligogenic editing could be more of a benefit or harm. Moreover, oligogenic editing is far more likely to be socially acceptable than polygenic editing (which I think is similarly unlikely to be socially accepted as ESPS of 10,000s of embryos).

We thank the referee for the interesting and proposed separation between the effects of “oligogenic” (= good or at least might be) versus “polygenic” (= bad) gene editing. We do now discuss this suggestion in the paper (lines 572-579) but have not changed the overall nomenclature for a number of reasons. First, we find it very difficult to predict with any certainty the level of maturity and safety of HPE technologies over a 30-year horizon. Second, as acknowledged by the referee, there is no clear line between “oligogenic” and “polygenic”, such that advocating for one but not the other might become a moving target. Third, not all diseases have rare protective variants of large effects, so that editing many loci might be the only way to achieve a given change. Fourth, it is possible, perhaps likely, that the deleterious effects on fitness are proportional to the predicted/achieved change and not the number of variants required to achieve this. Fifth, as stated by reviewer #3, it is possible that rare protective variants have deleterious epistatic effects that will be revealed when they are homozygous. Sixth, all of the common diseases and their risk factors we have modelled are considered complex “polygenic” traits, and “oligogenic gene editing for polygenic traits” may lead to confusion. We have tried to capture these points in a paragraph in the Concluding Remarks section (lines 572-579).

Some other issues relating to the statistical genetic modelling and results:

- For the main results presented, the authors chose not to model the more realistic scenario that only a very small fraction of the population would perform PGE. If there was an e.g. 1% uptake (perhaps on this high side), then there would be almost no change in the population prevalence of these diseases, and although results/text refers to ‘edited genomes’, I think this point should be made crystal clear to the readers so that they understand the likely (lack of) impact of future genome editing on common diseases in populations (unless rolled out by national healthcare systems, which seems unlikely).

We agree and have made this clear (lines 105, 108-109, 125, 151-155).

- For the disease plot of Fig.1, the Y-axis should show fold reductions in increments of 1 between 1x and 10x (2x, 3x, 4x, etc) since these are where most of the ‘action is’ here but the reader will be unable to easily estimate what fold changes these correspond to.

We thank the reviewer for this suggestion to improve clarity and have now shown the y-axis in units of 2, 4, 6, 8, 10, 20, 30, 40, 50 and 60-fold reduction.

- The new section at the top of “Model limitations..” about potential advances in somatic editing obviating the need for heritable editing, seems out of place here and would surely be better suited to the introduction or at the end of the paper. The link here is with ‘future environments’ as though addressing G*E effects, but this link seems highly tenuous.

We thank the referee to this comment and have created a new paragraph with a 7th limitation (lines 308-325).

- The modelling of G*E effects, as presented in Fig.2, is rather simplistic in relating to assuming that future environments will simply change traits by some degree. More pertinent G*E effects are those that could occur after dramatic changes to traits, given the balance created between observed genetic variation and the environment caused by selection (i.e. G*E effects with highly deleterious results due to genetic profiles poorly matched to the present environment), or those that mean that diseases such as MDD may operate in the population in a relative rather than absolute sense (i.e. prevalence remains stable, despite the inferences of Fig.1, due to prevalence being more due to relative risk than absolute risk). I think the possibility of such G*E effects should be at least mentioned here.

We have now mentioned other GxE effects such as proposed by the reviewer (lines 188-191).

- L258 “the effects inducing large changes associated with most polygenic traits are largely unknown” - there is now a sizeable literature suggesting that most polygenic traits have been under stabilising selection, which would indicate that large changes in their values would typically result in deleterious effects. The authors should make the readers aware of this. While the authors model stabilising selection as presented in Fig.3, they do not explain that evidence suggests that most polygenic traits are under stabilising selection and that this likely restricts the extent of editing that is viable and consistent with health in the present environment ('present' on the scale of centuries).

We agree that there is strong evidence of negative selection and now cited relevant literature (lines 253-256).

- When the authors refer to 'reduced fitness' I think best to emphasise (for more lay readers) that they're referring to large-scale suffering and death, since this is how the consequences of present complex diseases are described (justifiably) when highlighting the potential benefits of editing.

We have now defined fitness in this context as disease-associated reduction in fertility/viability and disease-associated increase in mortality and morbidity (lines 205-206).

- Enhancement section: as raised previously, given scientific debate about the meaning of 'IQ' and about indirect (/household) and non-cognitive effects in relation to Education GWAS, and the links between 'IQ' and past eugenics, I'd highly recommend dropping or at least limiting references to these in the paper. References to human 'enhancements' (also in the abstract), in relation to IQ and height (despite being dropped from the modelling), seem ill-judged - how can 'enhancements' of such traits be defined in the context of evident stabilising selection and pleiotropic effects (and given personal subjectivity in terms of ethics)? (unless by enhancements the authors mean edits that make individuals closer to the average).

We have now limited the references to “enhancement” and use a more neutral “non-disease traits” instead. In the light of comments from reviewer #3 we have clarified that although this term is used in the reproductive ethics literature it has problems of definition. We believe it would be wrong not to at least mention gene editing applied to non-disease traits (“enhancement”) because it is a topic actively discussed in the (bio)ethics literature and at major gene editing conferences. (lines 366-384)

- While Fig.3 indicates how inequality due to editing changes with uptake, it shows this in relative terms, which obscures the degree of inequality itself. Also, the prevalences (K) and impacts of editing (K') are for hypothesised diseases. I think it could be even more informative to readers (at least as a replacement of one of the 4 plots shown) to show this relationship in terms of absolute inequality of disease risk for the real diseases of Fig.1 based on the effects of editing 10 variants for each disease - this would show the degree of inequality (e.g. as a variance of risks) across uptake and would also show the relative inequality across the different diseases (not observable when standardised as now).

We thank the referee for this interesting suggestion and now show the results for the same 5 diseases as in Fig. 1 (SCZ, AD, CAD, T2D, MDD). We moved the previous Figure to the supplementary materials (Supplementary Figure 4).

Referee #3 (Remarks to the Author):

This perspective has been much improved by the omission of height and BMI as example traits and the increased focus on possible deleterious consequences of embryonic gene editing. The transparency of the model has also been improved by addition of source code and additional methodological detail. Overall, the work still seems pretty likely to spur controversy, but perhaps a more productive controversy than would have been the case in its original form.

We thank the referee again for their time spent on reviewing our paper, for acknowledging improvement and for providing additional constructive feedback. We have made further changes to address the reviewer's comments, made textual changes to improve clarity and changed the main acronym to HPE (heritable polygenic editing) to avoid confusion with polygenic prediction acronyms and pre-implantation testing acronyms.

There are still some places where the article expresses a viewpoint that feels dangerously close to eugenics: the view that society should collectively decide to eradicate certain undesirable genes and encourage treatments that will nudge society toward a genetic profile that is optimized for producing some notion of a good life. I think this problematic stance could be tempered by adding additional emphasis on the role of individual choice: in my view, scientists and politicians have a responsibility to decide that certain gene editing treatments should never be developed because they would have a harmful effect on society, but the converse is not true: they do not have the right to recommend that certain gene edits should be broadly adopted. We see this distinction with the way that prenatal genetic testing is offered to patients today: scientists and doctors have decided that this testing is enough of a social good that it should be routinely offered to patients, but it would not be ethical for anyone's doctor to pressure them to consent to such testing.

We agree that any future uses of HPE must respect individual rights to autonomy, and should not impose state values on individuals. We now make this point more explicitly in the paper. We state:

"In order to be properly distinguished from previous eugenic movements, any use of HPE must respect individual reproductive liberty. The state should not impose an ideal of what constitutes a good life on individuals and should not use coercive measures to pressure people to use HPE. Rather, any future use of HPE should be modelled on modern clinical genetics, which uses genetic technologies to further the goals of medicine." (lines 346-350)

I think that the three paragraphs starting on line 473 get a bit too close to implying that people should one day be pressured to consent to gene editing in the way that doctors currently pressure people to quit smoking and lose weight. It's not clear how far the authors are thinking we should go in having "substantial ethical reasons to use the technology to prevent the transmission of these genes," so it would be best to clarify that we should only go as far as offering such editing to enthusiastic customers.

We have now deleted these paragraphs. We agree with the reviewer that there was some ambiguity in these paragraphs, and it would take much more space to properly discuss if/when we would have substantial ethical reasons to use HPE. We have made changes throughout the manuscript to clarify that our goal is to draw attention to these ethical issues and start a discussion, rather than arrive at any conclusive positions.

Another critique of this section is that it could be problematic to use "overall risk of death" as

a criterion for offering gene editing. After all, a gene associated with darker skin or non-conforming gender identity probably increases the risk of death in a society that is biased against these traits, but that does not seem like a good reason to encourage editing of such loci.

We have now deleted this section, and do not refer to overall risk of death.

The role of individual choice and agency should also be added to the section “Moving forward—the need for collectivist ethical approach to human genome editing” to emphasize where collectivist imperatives end and become trumped by individual choice. Specifically, individuals have the ultimate deciding power about what they think constitutes a good life and whether that makes them want to avail themselves of currently available gene editing technologies.

I take the authors’ point that the term “human enhancement” has precedent, but it still seems like a more neutral term like “attempts at human enhancement” or “cosmetic alterations” might be preferable to emphasize the subjective nature of such efforts.

We thank the referee for this comment. The term “human enhancement” has been used in many ways throughout the bioethics literature. The main risk we want to draw attention to is the potential of HPE to be used to target non-disease traits which is often referred to as “enhancement” in the bioethics literature. We now make explicit in the text that we are using enhancement to refer to its use for non-health related purposes. The edited section reads:

“The same techniques that enable HPE to reduce risk of disease could also be used to alter non-disease traits, including physical attributes, personality and cognitive traits. Using genetic technologies for purposes other than treating diseases, sometimes called human “enhancement” in the bioethics literature⁵⁶, raises specific ethical concerns. We note that the term “human enhancement” is highly contested and that a more neutral term such as “change in non-disease traits” is preferable.” (lines 367-370)

The discussion of SES is overall pretty thought provoking and nuanced. One additional point that occurred to me is that low-SES individuals might feel more pressure than high-SES individuals to avail themselves of gene editing technologies in the same way that they can feel extra pressure to strong-arm their kids into studying hard and pursuing high-earning careers.

We agree that such pressures exist and that they can apply to a range of choices for individuals and their families.

I found the discussion of pleiotropy to be much improved, but one important thing that is not mentioned is that rare protective alleles are especially likely to have epistatic consequences we are not aware of since we have not had a chance to observe them in combination with very many other rare alleles, and natural selection has also not had the chance to select against any deleterious effects that these alleles might have in combination with various other rare variants. It seems likely that any attempt to make rare alleles more common should be preceded by some kind of variant effect scan that attempts to test how these alleles behave on a variety of backgrounds and anticipate deleterious epistatic effects.

We thank the reviewer for the interesting point about rare protective variants and have now explicitly mentioned this in the text (lines 278-280).

Referee #4 (Remarks to the Author):

As a result of your responses to the reviewers, this paper has become both stronger (more publishable) and weaker (in its implications). And that does seem to me the right result.

We thank the referee again for their time spent on reviewing our paper, for acknowledging improvement and for providing additional constructive feedback. We have made further changes to address the reviewer's comments, made textual changes to improve clarity and changed the main acronym to HPE (heritable polygenic gene editing) to avoid confusion with polygenic prediction acronyms and pre-implantation testing acronyms.

Assuming the scientific reviewers accept your changes (the validity of which I am in no position to judge), I think it would be a useful contribution to the literature. I do, of course, have some comments. Three of them are general but most range from small to trivial (typos). I'll start with the big ones.

General Comments

First, I think your responses to the reviewers have introduced substantial uncertainty into the question of the power of PGS and to its risks. This seems to me necessary, but I do think entails some more changes in the article. The current version is not quite as strongly "this will happen" as the earlier version, but I think its overall tone does not yet reflect the increased uncertainties of its text.

We have made substantial edits throughout the manuscript to emphasize the uncertainties involved and to provide a more cautious picture of the future role of HPE in reproduction. Some specific changes made are noted in response to the comments below.

For example, in your concluding comments you say "over the next decades, it will become possible to" That will, at least to the extent it implies "safely and effectively", needs to be weakened—to "it may" or "it is plausible that" or even (though I personally wouldn't say this) "it is likely that." But not "it will."

We have revised as suggested. The new sentence reads:

"over the next decades, it may become possible to make multiple edits of the DNA sequence of human embryos and germ cells." (lines 565-566).

I think you should explicitly, and up front, state that whether PGS will make sense in light of the uncertainties about its advantages and risks remains unclear. But because it is at least plausible (possible?) that it will work and its effects may be very large, it requires attention well in advance of efforts to use it.

In addition to making changes to the abstract, introduction and conclusion, to give the paper a more cautious tone, we have added the following sentences to explicitly acknowledge that the uncertainty regarding HPE's use:

"Whether multiplex editing of polygenic traits becomes practical or desirable, given the balance of risks and benefits, is highly uncertain and will depend in part on as yet unknown safety and efficiency considerations. However, it is a prospect that it is worth taking seriously, given the potentially disruptive and transformative consequences. The social and ethical structures in place should HPE become available will have a profound impact on future generations." (lines 79-84)

Second, it occurred to me this time that your foreseen world, with widespread adoption of PGE, will require the almost complete abandonment of natural conception. I have myself previously argued that in vitro gametogenesis plus whole genome sequencing will lead a large percentage of the population to avoid monogenic conditions by using PGT (preimplantation genetic testing, in this context – your use of PG for polygenic could lead to some reader confusion). But that prediction encountered a lot of skepticism. I think the needed very wide transition to clinic conception should be noted somewhere in this.

We agree that HPE will likely require the use of assisted reproduction technologies (e.g., IVF), and that this is a barrier to the widespread use of HPE, and its capacity to substantially reduce population-level prevalence of polygenic diseases. The modelling we have conducted refers to change among “edited genomes”, and we think the changes modelled among edited genomes are themselves significant enough to raise ethical issues. We have made changes throughout the manuscript to make it clearer that the changes modelled refer to those with edited genomes and not to population prevalence.

In addition, we now explicitly state that a limitation of the model is that it is not predictive of changes in population-level disease prevalences, and that dramatic changes to population prevalence in a single generation is unlikely.

“Sixth, while we model changes among those with edited genomes, which are not predictive of changes in population-level disease prevalence. HPE would only be capable of drastically altering population prevalence if a large proportion of the next generation is born through HPE. As HPE will likely only be possible through the use of in-vitro fertilization, and thus incompatible with reproduction via sexual intercourse, it may be unlikely that it will have large effects on population prevalence in the absence of other advances which make HPE compatible with natural reproduction (such as the ability to modify spermatogonial stem cells).” (lines 297-303)

Third, you end by saying “we” must use the time to figure things out. But who is the “we”? I’m not asking for a roadmap but something along the lines of discussions leading to public debates and ultimately to governmental (?) decisions, at the national and (or?) international level. (A line noting the difficulties in that, especially internationally, would not be out of place.)

We have modified our conclusion to take account of these points. The relevant section reads:

“It is vital for national governments, as well as the international community, to carefully consider how to regulate HPE in order to best manage the ethical challenges. In doing so, it is important to also consider the risks of deciding not to use HPE. Polygenic diseases are a leading cause of premature death in the world, strain the health system, and reduce people’s freedom by making them reliant on medical resources. Successful management of the risks posed by HPE will likely require strong international cooperation, which is particularly challenging in the face of competing interests, national priorities and conflicting values. There are good reasons to start discussing the challenges and opportunities that HPE provides now, well before it may become a practical possibility. In any discussions about the potential use of gene editing to reduce genomic contribution to common disease or traits, it is important to quantify what might be achievable. We hope that our examples provide a foundation for informed and balanced discussion.” (lines 581-593)

Smaller Points

45-50 The discussion of Aurea seems awkward. You may need a half sentence to say why she is relevant to your discussion.

We have cut the whole paragraph on Aurea and replaced it with a sentence at the end of the preceding paragraph discussing He Jiankui. The new sentence reads:

“Lulu and Nana’s births have been followed by the birth of Aurea, the first child to be born via embryo screening using polygenic risk scores (ESPS).” (lines 42-43)

70 You are using PGS again (which I don’t like because of the possible confusion with PGT, PGD, etc.), but what happened to ESPS? Wasn’t that the new acronym/term you made up for your suggested process?

We apologise for the confusion but are trying to follow recent nomenclature in the literature. PGS are polygenic scores (sometimes also called PRS for “polygenic risk score” and PGI for “polygenic index”). They are DNA-based predictors of a phenotype and can be used in the context of personalised medicine, for example to identify individuals at higher risk of disease. They can also be used in the context of embryo screening and selection, and the approach of embryo selection based upon polygenic scores has been termed ESPS in the literature.

82 It IS possible to do functional test for phenotypes in vitro or it may, in some circumstances, be? Doesn’t that depend on the phenotype? If it is “protein X is being produced at level Y” it seems possible (though it may not occur in a full organism because of effects from other cells) but if it is, say, AD, you can’t. I’d weaken the statement.

We have qualified this statement to read:

“Furthermore, it is possible to test the functional effects of variants on protein expression in vitro, using tools such as experimental genome editing” (lines 72-73)

120 This line is ambiguous. I’m not sure what you mean by “reducing the lifetime prevalence by 5 to 2.9%.” You seem to be saying total prevalence, but I wonder about that, as lifetime prevalence of 5% AD seems awfully low. I suppose you could be saying “reducing the prevalence by somewhere between 5 and 2.9 percentage points” (from, say 15%) or even reducing the prevalence by 5 to 2.9 percent, which, with a 15% lifetime prevalence, would only take it down to 14.25 to 14.5%. I doubt you mean the latter two, but the sentence isn’t clear.

Also, does this involve editing out APOE4 alleles (possible certainly with heterozygotes) or replacing them with APOE2 alleles? And maybe replacing APOE3 alleles – because if you want to lower AD risk, you’d get rid of those, too. More clarity may be in the supplemental materials but a statement in text would help.

We thank the reviewer for this query. What we meant by that statement is that if we assume that the current lifetime prevalence is 5% then the predicted lifetime prevalence among edited genomes is 2.9%, so a proportional reduction of 42% ($1 - 2.9/5.0$). We have now clarified that sentence. We use values from the literature on the population lifetime prevalence for each disease we discuss. However, we acknowledge that such estimates are population/country dependent and that lifetime risk for AD is sex and age dependent and may change in ageing populations. In a study from the Framingham Study, the lifetime risk for a 65-year old man was estimated to be 6.3% (Seshadri et al. 1997, Neurology). In the UK Biobank, the

reported proportion of paternal AD cases (potentially inflated by other dementias) was 5.5% (Marioni et al. 2018, Translational Psychiatry).

129 Of course, the long-term effects of switching from common alleles to rare protective ones isn't well understood because we rarely have much data on the lifetime effects of the rare alleles on people. Even the CCR5 knock out effects are based on awfully limited data.

We agree with the referee and now explicitly mention this in the context of epistatic effects, following a similar comment from referee #3 (lines 278-280 and 578-579).

136 The "average genome" of "the population" – which population? This average genome will be different in different ethnicities, which should lead (I think) to different degrees of reduction in different populations, no? You do come back and make a point about different population at 279 but I think you should note it earlier.

We thank the referee for this comment. We defined "population" in this sentence by its allele frequencies and agree that those differ among ancestries. We have now clarified that the predicted reduction in disease prevalence is population specific (lines 129-131).

175 It's not just gene therapies. More traditional treatments may also be effective, like the drugs that treatment 90% of cystic fibrosis patients now. It's not "one and done" like the gene editing (somatic or germline) would be, but whether one treatment or the other is "better" depends on lots of things including risks, cost, and convenience.

We agree and did not wish to imply that it is only somatic gene therapies that may reduce the need for germline HPE. We have revised these lines accordingly which now read:

"We cannot predict future environments but one possible change of environment in the context of disease is new treatments and therapeutics that would obviate the justification for heritable gene editing." (lines 174-176)

177 Since you wrote this, the first CRISPR therapy approval HAS occurred in 2023, in the UK.

Indeed, CRISPR therapies have also recently be granted FDA approval and approval in the UK. We have revised this sentence and the relevant reference accordingly.

"first CRISPR therapy for sickle cell disease has been granted approval by UK and US regulators (ref 29)" (lines 212-213)

181 Typo – change "if" to "it"

Changed, thank you.

232 Is it "reasonable to assume"? Personally, I'd say "not unreasonable to assume"...a touch weaker. We know so little.

This has been changed as suggested.

246 Define or describe winner's curse – you use it only once, without explanation.

Done (line 237).

264 It would be good to highlight the huge uncertainty, and difficulties of understanding possible prenatal effects on gene changes.

We have added the sentence:

“It is very difficult to prospectively predict the pleiotropic effect of novel combinations of variants on prenatal development, and this will be a significant source of uncertainty in any future use of HPE.” (lines 247-249)

289 typo – edited for editing

Changed – thank you.

293 typo -smaller for small. Also, I don't really understand your fetal hemoglobin example. I don't understand why the treatment has such a big effect and but the allele has a small effect. Is the changed allele extremely rare (or even non-existent) in human populations? Whatever the truth, the example doesn't make sense to me (and so might not make sense to other readers).

We have fixed the typo and expanded upon the explanation.

306 Well, some of Europe did (though not the southern and predominately catholic parts), some Canadian provinces did, and there were efforts (though I think no government action) in Australia.

The new sentence reads

“Intellectual disability, psychiatric diseases, criminality and poverty were targets of the eugenics movements in Germany, other parts of Europe, Canada, Australia and the US from the late 19th to early twentieth century” (lines 337-340)

313 Well, it's still to control what kind of people exist, but with the decision at the family level and not the societal one. (Although there may be some tension in you putting emphasis on this in a paper where you end by calling for more collective thinking, which some might conclude should include encouraging or requiring this kind of editing for the collective good.)

We have changed the wording and the paragraph now reads:

“In order to be properly distinguished from eugenic movements, any use of HPE must respect individual reproductive liberty. The state should not impose an ideal of what constitutes a good life on individuals and should not use coercive measures to pressure people to use HPE. Rather, any future use of HPE should be modelled on modern clinical genetics, which uses genetic technologies to further the goals of medicine. Clinical genetics in democratic societies is based on non-directive counselling, provision of information and choice, interventions aimed at the well-being of the future child and is voluntary. When implemented with appropriate regulation and governance, HPE can be distinguished from eugenics movements just as contemporary clinical genetics is.” (lines 346-354)

In addition, we have changed the section on collective considerations to stress that although we need collective thinking to define what the goals of using PGE should be, any future use should respect basic right of autonomy. This new section reads:

“While collectivist considerations should inform the values of governments, and the goals they pursue, it is also important that those goals not override basic human rights, such as the right to autonomy. While the goals and aims of governments should be informed by collectivist considerations about what is a good for society as a whole, and what a good human future looks like, respect for human right must come first.” (lines 535-539)

Ethics Box

Why not question whether new phenotypes would be bad – are all differences are bad? If you are listing counterarguments, I’d make that one.

We have added a safety heading to the safety box, to address what we think is the reviewer’s concern. The added section reads:

Safety HPE introduces novel combinations of variants, which could be dangerous and unsafe. It would be unethical to impose this uncertain risk on future generations	It is vital that any use of HPE be supported by rigorous safety data and have a clear justification through risks/benefit balance. Natural reproduction also generates new combinations of variants. One strategy is to limit HPE to introducing variant combinations already seen in natural populations.
--	--

Typo in “means matter” – overlook for overlooked

Changed – thank you.

335-6 Hmm, height and IQ show up. I thought you took them out. Seems like you are trying to sneak them back in but through a side door, without actually “showing your work.” I’d either leave them out entirely or mention them only a weak way: “It is even conceivable that, at some point in the future, traits like height and intelligence might be subject to large scale changes.

We have deleted the sentence that was there and added this sentence:

“It is conceivable that, at some point in the future, HPE could be used to target traits like height and intelligence, and lead to large scale changes in these traits” (lines 376-377)

In general, we believe it would be wrong not to at least mention gene editing applied to non-disease traits (“enhancement”) because it is a topic actively discussed in the (bio)ethics literature and at major gene editing conferences (see also comment to referee #2). We have tried to temper the language and tried to avoid using “enhancement”, even though that word is in the title of a number of papers we cite.

350 You talk about “concerns” and take them seriously, but “concerns” aren’t necessarily “arguments.” Taking them seriously politically may be necessary but do they deserve to be taken seriously intellectually without more?

We believe that for the purpose of this article, it is sufficient to note that many members of the public draw a sharp distinction between therapy and “enhancement”, and that this should be taken politically seriously, especially in a collectivist approach. There are ways in which this distinction might be drawn where it should

also be taken intellectually seriously, but we do not have space to explore this in more detail.

372 This seems to me repetitive of the sentence at 369.

We have revised this paragraph in response to comment from another reviewer and this section no longer repeats.

416 You might want to refer to differential health risks already by SES or ethnicity, such as huge black/white health disparities in the U.S.

We agree and have now added a reference to socioeconomic disparities in health in the US. (line 455)

434 Equitable access “to new technologies, such as GWAS today”? What access is currently needed to the “technology” of GWAS? I suspect you’ve got something specific in mind; if so, say it.

We agree that this was awkwardly worded and have now tried to clarify what we mean and provided a reference to health disparities and GWAS. (lines 473-474)

449 The “it depends on how it’s regulated” take seems inconsistent with the problems of international disparities, which you highlighted four lines earlier. Fixing that would take a lot more than good regulation or non-market provision. You might acknowledge the special difficulty there.

We have changed the way we end this sentence to remove reference to regulation. The revised section reads:

“There will be no easy solutions to these problems, which is why it is vital we start to consider the implications now, while the prospect of HPE is still many years away.” (lines 489-490)

453 I wouldn’t use the heading here “Pleiotropy and Genetic Diversity.” Table 1 mentions diversity (VERY briefly) but not pleiotropy. I think this header might confuse a reader, if only momentarily.

We have changed to heading to just “Diversity”, because the main risks of pleiotropy were already discussed in an earlier section. We have rewritten the Diversity section to focus on concerns about genetic diversity that are distinct from pleiotropy. (lines 492-508)

473 This paragraph seems very different from the rest of the paper and feels out of place. Maybe it needs some more explanation to fit it into your flow. BTW, is there much work on PRS for overall risk of death? If so, it might be good to mention it.

We have deleted this paragraph now in responses to comments by another reviewer

505 typo – that do not

Fixed.

510 “Genetic load”. You state, without citation, that health care is causing mildly deleterious mutations [do you mean that, or alleles?] to accumulate.” I believe that must be the case but is there evidence that it has actually happened? I presume the models referred to in the next

sentence are mathematical models, not based on actual data about known increases in, say, type 1 diabetes predisposing alleles.

We thank the referee for this question. We now cite the paper from Lynch (2016) in this sentence (we had cited this paper subsequently and qualified the statement to say “deleterious mutations are predicted to accumulate” (line 526). Lynch makes a strong logical case that the increasing genetic load is an inevitable outcome from knowledge of the germline mutation rate in humans and that clinical procedures for mitigating the consequences of deleterious mutations (for example surgery, medication, nutritional supplements, physical and psychiatric therapies) must result in the relaxation of natural selection against a range of deleterious mutations.

On this whole argument, even if the genetic load argument is right, those future generations could presumably make up their own minds about ways to improve their health, through your PGE, through PDT, or otherwise. Arguably you are doing them a disservice by limiting their choices. (Assuming they will have choices, which I think you do.)

We thank the reviewer for this comment. One problem is that by the time individuals have the capacity to choose to have HPE for themselves, it will no longer be possible, as germline changes can only be made at the germ cell or embryonic stage. Although somatic editing could be performed at the time of making a personal choice, this is likely to be less efficient (as you need to target millions of cells) and does not provide protection for future generations. We have added a new section where we discuss the possibility that somatic cell therapies will reduce the desirability of HPE (308 – 328).

Our goal is to fuel and enlighten a discussion on the potential benefits and risks of HPE, not to make any decision for future generations. The question is therefore how to ensure those decisions are well-informed and equitable.

520: I know this whole new section responds to reviewer comments. You could tailgate here on the National Academies’ group’s ideas on a path toward what it calls HGGE...in fact, you are recommending that the efforts start with HGGE.

We thank the reviewer for this suggestion which we have seriously considered. Although the translational pathway published by the National Academies “International Commission on the Clinical Use of Human Germline Genome Editing” is similar to our own, there are important differences which mean it is not suitable for us to tailgate off it. Most importantly, the Commission’s pathway takes lack of clinical alternatives to be the most important criteria when selecting initial targets for human heritable gene editing, whereas we take it be “expected benefit”. The Commission’s category A (initial targets for human heritable gene editing) includes editing to prevent adult-onset disease like Huntington’s disease, and treatable conditions like cystic fibrosis, in cases where PGD is not feasible. We think these conditions should only be targeted after success in using gene editing for lethal, untreatable, childhood onset disease. We think explaining these differences would take us too far off topic in this article.

522: The sentence makes it sound like Tay-Sachs editing would be PGE, but it’s really HGGE. I’d say “We should first evaluate the use of embryo editing, sometimes called human germline genome editing [if you don’t use that or something else for it earlier] for”. Tay-Sachs.

Similarly at 532, the “it” that could be extended isn’t really PGE, just HGGE. Basically, this would be a lot clearer if you just said that PGE’s pathway would necessarily follow on, and after, success with the HGGE pathway.

And, of course, first would really be preclinical work, in vitro with human embryos and in vitro and in vivo with non-human animals, especially non-human primates.

We have changed the paragraph to take account on these comments. The relevant section now reads:

“Before any human uses of heritable gene editing are considered, it should first be shown to be safe and effective in animal models, particularly non-human primates. The first human uses of genome editing should be performed in lethal. single gene disorders such as Tay Sach’s Disease.” (lines 542-545)

And to the extent you want to use rare protective alleles (which you clearly do), I think you should call for more research on the consequences of those rare alleles on total lifespan health of people who naturally carry them.

We now say,

“Before such applications are considered, it will be important to conduct more research on the effect of polygenic variants on individuals in natural populations, including the lifelong consequences of carrying rare protective alleles.” (lines 556-559)

540 You say “children” screened at birth but I think it’s still just “at least on child.”

We now say,

“Advances in technology³² have already resulted in the birth of at least two gene-edited children” (line 564)”

546 Once again, you sneak back to height and IQ. See my comment at 335.

We have now deleted reference to IQ here.

I think that’s it – more than enough, I’m sure you feel. Obviously, some of these are more important than others, especially on issues of style. I hope this helpful. I do think —if this passes muster with the scientific reviewers —this would be a useful article.

We thank the referee again for the many helpful queries, comments and suggestions.

Response to remaining referee comments

The authors wish to thank all reviewers again for their many comments and suggestions to improve the manuscript.

Referee #2 (Remarks to the Author):

This manuscript is now vastly improved from the original. I am satisfied with the latest round of responses and any remaining points of disagreement are essentially subjective/stylistic in nature (in particular about it being most worthwhile to focus on and discuss the editing of a small number of variants, rather than on more 'polygenic' editing, which seems to me worth effectively ruling out in the short/medium term).

One point though - line 20 in the abstract reads "Editing a relatively small number of genomic variants can make a substantial difference to an individual's risk.." - should change "can" to "could" since it is not known for certain (at least not from modelling).

We thank the reviewer and have made the requested change.

Referee #3 (Remarks to the Author):

The manuscript is much improved--at this point, it seems likely to spur healthy debate rather than unhealthy amounts of controversy. The authors have done a great job incorporating all of our diverse feedback while still making concrete quantitative points that will add to the discussion on this topic.

We thank the reviewer and also hope that our paper will generate debate rather than controversy.

Referee #4 (Remarks to the Author):

So, authors, have you second-guessed during this process whether you should have ever started this paper? I want to compliment you for sticking with it and for responding so well to the many comments made by your reviewers. Your responses seem to me to range from convincing to adequate (meaning "ok but not the way I would have said it"). Overall, I think the comments from the reviewers (especially the scientific reviewers) AND your responses to them have improved this paper markedly. And I think this is (almost) ready to be published.

We thank the reviewer for this comment and indeed asked ourselves the question raised on multiple occasions.

I have only one last (two pointed) comment, which I had earlier tagged to line 520, about the National Academies/Royal Society efforts. I accept your argument for why your approach is importantly different, and, with understanding but some regret, your view that explaining the differences would be too distracting.

Point one - think again about whether explaining the differences would take you too far afield. In the responses to the review, it took you only about 10 lines to explain them well. I bet you could get it down to 5 or 6 and I think it would worthwhile...as at least some, sophisticated, readers will say, "what about the NASEM/RS approach, which they don't even deign to mention. Did they not read it or are they avoiding it for other reasons?"

Point two - even if you don't talk about the NASEM/RS approach, you should at least cite it somehow, if only to make it clear that you DID notice it. (Plus, it's polite...and, note, I wasn't involved in that report at all, as far as I can recall.)

In response to both these points, we have now included a new paragraph that discusses and cites the report:

An international commission from The National Academies of Sciences, Engineering and Medicine produced guidelines in a study report “Heritable Human Genome Editing”⁷. The commission’s pathway takes lack of clinical alternatives to be the most important criterion when selecting initial targets for human heritable gene editing, whereas we propose “expected benefit”. The commission’s category A (initial targets for human heritable gene editing) includes editing to prevent adult-onset disease like Huntington’s Disease, and treatable conditions like cystic fibrosis (in cases where pre-implantation genetic diagnosis is not feasible). In contrast, we propose these conditions should only be targeted after success in using gene editing for lethal, untreatable, childhood onset disease⁴⁶.

Otherwise, a very nicely done job.